# Innate immune and proinflammatory signals activate the Hippo pathway via a Tak1-STRIPAK-Tao axis

Yinan Yang[1,4], Huijing Zhou[1,4], Xiawei Huang[1,4], Chengfang Wu[1], Kewei Zheng[1], Jingrong Deng[1], Yonggang Zheng [2], Jiahui Wang[1], Xiaofeng Chi[1], Xianjue Ma [3], Huimin Pan[1], Rui Shen[1], Duojia Pan [2] & Bo Liu [1]✉

The Hippo pathway controls developmental, homeostatic and regenerative tissue growth, and is frequently dysregulated in various diseases. Although this pathway can be activated by innate immune/inflammatory stimuli, the underlying mechanism is not fully understood. Here, we identify a conserved signaling cascade that leads to Hippo pathway activation by innate immune/inflammatory signals. We show that Tak1, a key kinase in innate immune/inflammatory signaling, activates the Hippo pathway by inducing the lysosomal degradation of Cka, an essential subunit of the STRIPAK PP2A complex that suppresses Hippo signaling. Suppression of STRIPAK results in the activation of Hippo pathway through Tao-Hpo signaling. We further show that Tak1-mediated Hippo signaling is involved in processes ranging from cell death to phagocytosis and innate immune memory. Our findings thus reveal a molecular connection between innate immune/inflammatory signaling and the evolutionarily conserved Hippo pathway, thus contributing to our understanding of infectious, inflammatory and malignant diseases.

The innate immune and inflammatory response which can be triggered by various harmful stimuli such as pathogen toxins, toxic compounds or tissue injury, are important protective mechanisms of the body[1]. Tak1 (Transforming growth factor-β (TGF-β)-activated kinase 1), a member of the mitogen-activated protein kinase (MAPK) kinase kinase (MAP3K) family, plays pivotal roles in innate immune and inflammatory response[2,3]. Pathogen-derived molecules, such as lipopolysaccharides (LPS), or proinflammatory cytokines, such as tumor necrosis factor α (TNFα) and interleukin-1β (IL-1β), bind to their specific transmembrane receptors to elicit downstream signaling cascade resulting in Tak1 activation. Activated Tak1 in turn activates the inhibitor of κB kinases (IKKs) and MAPK kinases to trigger the nuclear factor-κB (NF-κB) and MAPK signaling, respectively[4–6]. Tak1 is well conserved in *Drosophila* and mediates the activation of immune

deficiency (Imd) pathway by Gram-negative bacterium infection[7,8] or Jun amino-terminal kinase (JNK) pathway by the TNFα family ligand Eiger[9,10].

Initially identified in *Drosophila*, the evolutionarily conserved Hippo pathway is a key regulator of tissue growth through coordinated regulation of cell proliferation and cell death in diverse animals[11–15]. Central to this pathway is a kinase cascade wherein the Hpo (MST1/2 in mammals)-Sav (SAV1 in mammals) kinase complex phosphorylates and activates the Wts (LATS1/2 in mammals)-Mats (MOB1A/B in mammals) kinase complex, which in turn phosphorylates the transcriptional coactivator Yki (YAP/TAZ in mammals). The phosphorylation of Yki S168 (YAP S127 and TAZ S89) creates a binding site for 14-3-3 proteins and excludes Yki/YAP/TAZ from the nucleus[16–18], where they normally function as coactivators for the TEAD family transcriptional

[1]State Key Laboratory of Cellular Stress Biology, Innovation Center for Cell Signaling Network, School of Life Sciences, Xiamen University, Xiamen, Fujian 361102, China. [2]Department of Physiology, Howard Hughes Medical Institute, University of Texas Southwestern Medical Center, Dallas, TX 75390, USA. [3]Westlake Laboratory of Life Sciences and Biomedicine, Hangzhou, Zhejiang 310024, China. [4]These authors contributed equally: Yinan Yang, Huijing Zhou, Xiawei Huang. ✉e-mail: bliu23@xmu.edu.cn

factor Sd (TEAD1/2/3/4 in mammals) to drive the expression of Hippo pathway target genes involved in cell proliferation and survival. Following the discovery of these core components, additional regulators have been identified to converge on Hpo/MST to regulate the Hippo pathway activity. The sterile 20-like kinase Tao-1 (TAOK1/2/3 in mammals) directly phosphorylates the key residue T195 within the activation loop of Hpo (T183/T180 for MST1/2) and activates Hpo/MST[19,20]. On the other hand, the Striatin-interacting phosphatase and kinase-protein phosphatase 2A (STRIPAK PP2A) complex dephosphorylates and inactivates Hpo/MST[21,22]. Although the Hippo pathway was initially identified as a central regulator of organ size, studies in the past decade has extended its physiological function to additional processes such as innate immunity.

Studies in *Drosophila* have provided significant mechanistic insights into the interplay between Hippo pathway and innate immunity. The innate immune response in *Drosophila* includes cellular reactions, which involve phagocytosis or encapsulation of the pathogens by hemocytes, and humoral reactions, which involve the generation of antimicrobial peptides (AMP) by fat bodies[23,24]. Plasmatocytes, the most abundant hemocytes in circulation, are considered equivalent to vertebrate macrophages. They possess phagocytic ability and are responsible for the engulfment and digestion of intruding pathogens. As for the humoral reactions, they are primarily mediated by two signaling pathways, namely Toll and Imd[25,26]. Toll pathway is mainly activated by Gram-positive bacteria and fungi, while Imd pathway is mainly activated by Gram-negative bacteria[24,27,28]. The Imd pathway closely resembles the mammalian TNF receptor (TNFR) pathway to regulate the NF-κB transcription factor Relish (Rel) which governs the expression of a battery of distinct AMP. Rel phosphorylation mediated by Tak1 is a prerequisite for Rel activation[8,29]. The Hippo pathway has been implicated in the regulation of the humoral immune response by functioning downstream of Toll signaling in *Drosophila*[30]. Whether Hippo pathway can be activated by Imd signaling in *Drosophila* remains unknown. Meanwhile, there is evidence suggesting that immune and inflammatory stimuli activate the Hippo signaling pathway in mammalian cells, but the underlying mechanism remains largely unclear[31–33].

In this study, we report the identification and characterization of Tak1 as a key factor mediating the activation of the Hippo signaling by innate immune/inflammatory stimuli both in *Drosophila* and in mammal. We show that activated Tak1 acts as a cargo receptor for the selective autophagy of the Cka subunit of the Hippo inhibitory STRIPAK PP2A complex, and induces the lysosomal degradation of Cka. Suppression of STRIPAK PP2A complex leads to Hippo signaling activation through the Tao-Hpo signaling axis. We further show that Tak1-mediated Hippo signaling is involved in cell death, phagocytosis and innate immune memory. Our findings thus unravel a link between the immune/inflammatory signaling and the Hippo signaling, shedding light on our understanding and therapeutics of infectious, inflammatory and malignant diseases.

## Results

### Identification of Tak1 as an activator of Hippo signaling

Our previous study identified a crosstalk between the Toll signaling pathway and the Hippo signaling pathway in *Drosophila*. In response to Gram-positive bacterium or fungus infection, the Toll-Myd88-Pelle cascade activates Hippo signaling through Pelle-mediated phosphorylation and degradation of Cka, which is the regulatory subunit of the Hpo phosphatase STRIPAK PP2A complex. However, whether the Imd pathway is able to activate Hippo signaling remains unknown. To address this question, we first treated *Drosophila* S2 cells with cell wall constituents of Gram-negative bacteria, LPS or peptidoglycan (PGN), and then examined the phosphorylation levels of Hpo and Hpo substrate Mats. As shown in Supplementary Fig. 1A, either LPS or *E. coli* PGN treatment markedly induced Hpo or Mats phosphorylation,

indicating activation of Hippo signaling. These results thus suggest that, like Toll pathway, activation of Imd pathway is also able to activate Hippo signaling in *Drosophila* cells.

To understand how Imd pathway activates Hippo signaling, we searched *Drosophila* Interaction Database (DroID, http://www.droidb.org), a database that assembles protein interaction data from a variety of sources[34], for protein interactions between components of the Imd pathway and the Hippo pathway. We found that only Tak1 and its binding partner Tab2 have been identified as interacting proteins of Hippo pathway. Of particular interest is Tak1, which has been found to associate with two subunits of the STRIPAK PP2A phosphatase complex in a genome-wide affinity purification coupled with mass spectrometry analysis (AP-MS)[35]. This result, together with our previous study showing that STRIPAK complex mediates the crosstalk between Toll pathway and Hippo pathway, prompted us to further investigate a potential role of Tak1 in mediating Hippo signaling activation by Imd pathway.

We first tested whether Tak1 is able to activate Hippo signaling. To this end, we co-transfected epitope-tagged Tak1 and core Hippo pathway components into *Drosophila* S2R+ cells, and examined the phosphorylation status of these components. Strikingly, the phosphorylation of all tested core Hippo pathway components including Hpo, Wts, Mats and Yki was induced by Tak1, but not the kinase-dead mutant Tak1[S176A][36] (Fig. 1A–D), suggesting that Tak1 can activate Hippo signaling and this process depends on the kinase activity of Tak1. Since it is known that Yki S168 phosphorylation relies on Wts and we found that Tak1 can induce Hpo and Wts phosphorylation, we reasoned that Tak1-induced Yki S168 phosphorylation depends on Hpo and Wts. Indeed, Tak1-induced Yki S168 phosphorylation was dramatically reduced by depletion of Hpo or Wts (Fig. 1E) (see Supplementary Table 1 for the knockdown efficiency of all RNAi reported in this paper).

Next, we explored how Tak1 induces Hpo phosphorylation. Tak1 is known to activate both the NF-κB and JNK pathways by functioning as the *Drosophila* IKK complex (IKKβ/*ird5* and IKKγ/*key*) and JNKK/*hep*-activating kinase, respectively[7,8,37]. To elucidate the underlying mechanisms by which Tak1 promotes Hpo phosphorylation, we first examined the effect of depleting downstream NF-κB and JNK pathway kinases on Tak1-induced Hpo phosphorylation. Interestingly, we found that Tak1-induced Hpo phosphorylation was not affected by knockdown of *ird5*, *key*, *hep* or *bsk* (Supplementary Fig. 1B), suggesting that Tak1-induced Hpo phosphorylation is independent of the downstream NF-κB and JNK pathway kinases.

We then examined whether Tak1 can directly phosphorylate Hpo by performing in vitro kinase assay using bacterially purified glutathione-S-transferase (GST)-tagged Hpo as substrate. Notably, we failed to detect direct phosphorylation of Hpo by Tak1 (Supplementary Fig. 1C). We therefore explored whether Tak1 promotes Hpo phosphorylation indirectly by phosphorylating a Hippo pathway component upstream of Hpo. Since Tao-1 is known to catalyze the phosphorylation of Hpo T195, we next tested whether Tak1-induced Hpo T195 phosphorylation is Tao-1-dependent. As such, we first examined whether Tak1 can promote Tao-1 phosphorylation. Interestingly, Tak1, but not Tak1[S176A], can induce Tao-1 phosphorylation (Fig. 1F). Moreover, Tak1-induced Hpo phosphorylation was largely diminished by simultaneous knockdown of Tao-1 (Fig. 1G). On the other hand, Tao-1-induced Hpo phosphorylation was potentiated by co-expression of Tak1 (Fig. 1H). These results indicate that Tak1-induced Hpo phosphorylation was Tao-1-dependent. In line with this notion, Tak1-induced Yki phosphorylation was suppressed by Tao-1 depletion (Fig. 1I).

Yki phosphorylation on S168 residue results in its cytoplasmic retention and inactivation[16]. Since Tak1 induces Yki S168 phosphorylation, it is conceivable that Tak1 is able to inhibit the transcription activity of Yki. To test this hypothesis, we performed

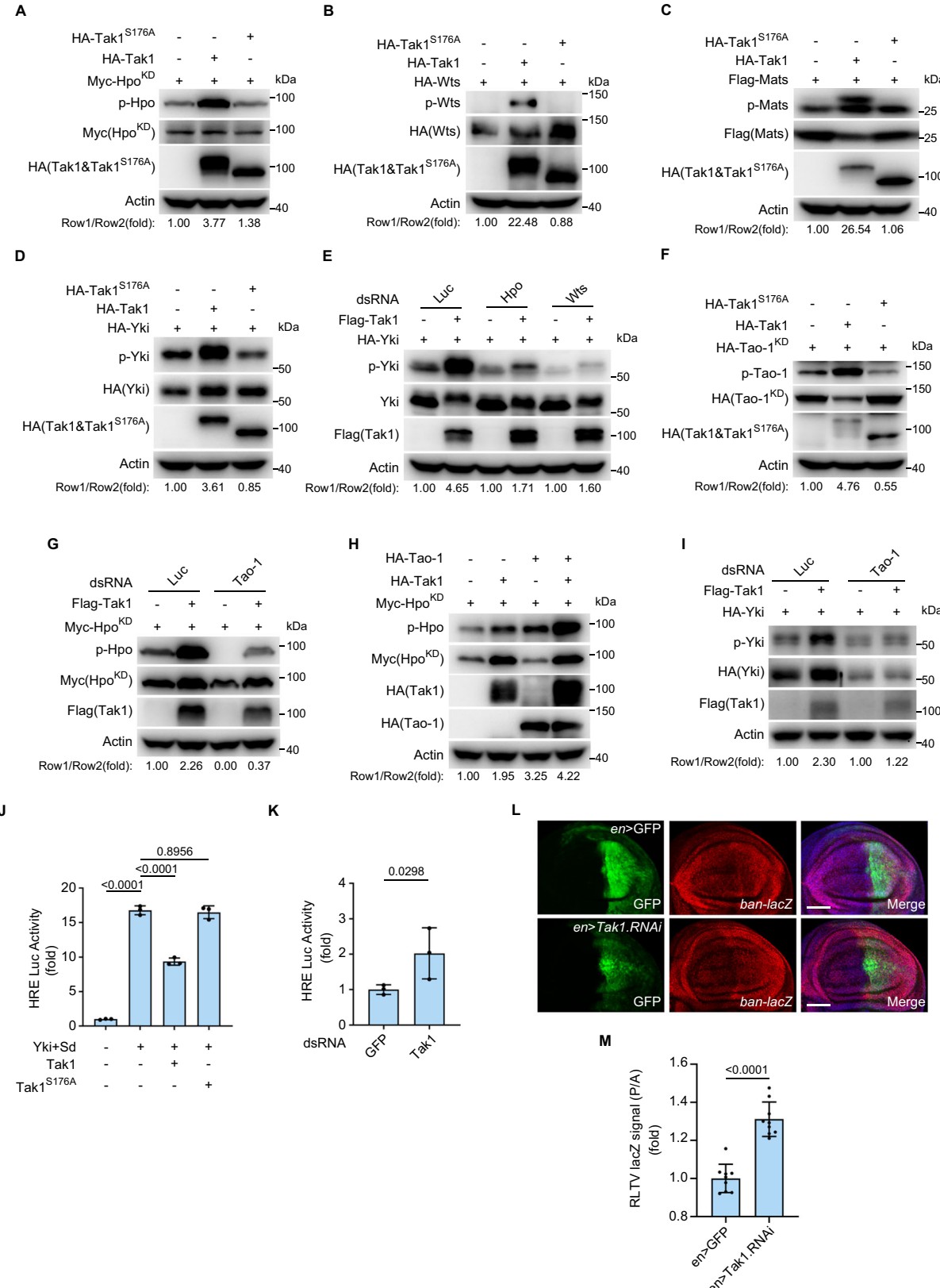

luciferase assay using a luciferase reporter driven by a minimal Hippo responsive element (HRE) from the Hippo target gene *diap1*[38]. Indeed, overexpression of Tak1, but not Tak1[S176A], potently reduced Yki/Sd-mediated HRE reporter activity in *Drosophila* S2R+ cells (Fig. 1J). On the contrary, knockdown of Tak1 significantly elevated the HRE reporter activity (Fig. 1K). To further corroborate the

inhibitory effect of Tak1 on Yki transcription activity in vivo, we examined the effect of manipulating Tak1 level on Yki activity in *Drosophila* wing imaginal disc using the widely-used *ban-lacZ* reporter of Yki activity. Knockdown of Tak1 driven by *engrailed*-Gal4 resulted in increased *ban-lacZ* level (Fig. 1L, M), suggesting enhanced Yki activity. Taken together, these results indicate that

**Fig. 1 | Tak1 induces the phosphorylation of Hippo pathway components and regulates Hippo signaling activity. A–D** *Drosophila* S2R+ cells were transfected with indicated plasmids. The phosphorylation of Hpo, Wts, Mats and Yki was detected via specific phosphor-antibodies against mammalian MST1/2 T183/T180 (**A**), Wts T1077 (**B**), mammalian MOB1 T35 (**C**) and Yki S168 (**D**). A kinase-dead form of Hpo (Hpo$^{KD}$) was used in (**A**) to avoid autophosphorylation of Hpo. **E** S2R+ cells were pretreated with dsRNAs of Luciferase, Hpo or Wts before transfected with indicated plasmids. Western blot was performed to monitor the phosphorylation of Yki. **F** Similar to (**A–D**) except for the transfected plasmids. The phosphorylation of Tao-1 was monitored via phosphor-antibody against mammalian TAOK2 S181. **G** S2R+ cells were pretreated with dsRNAs of Luciferase or Tao-1 before transfected with indicated plasmids. Western blot was performed to monitor the phosphorylation of Hpo. **H** S2R+ cells were transfected with indicated plasmids. Western blot was performed to monitor the phosphorylation of Hpo. Note the synergistic effect of Tak1 and Tao-1 on Hpo phosphorylation. **I** Similar to (**G**) except for the transfected plasmids. Note that the enhanced phosphorylation of Yki upon Tak1 co-expression was suppressed by Tao-1 knockdown. **J, K** HRE-Luciferase reporter was measured in triplicates in S2R+ cells transfected with indicated constructs (**J**) or pretreated with indicated dsRNAs (**K**). Data shown are mean ± s.d., $n = 3$ biological replicates. Data were analyzed using one-way analysis of variance (ANOVA), Tukey's honest significant difference (HSD) test (**J**) or two-tailed Student's $t$-test (**K**). **L, M** Third instar wing discs expressing UAS-GFP only (top panel) or UAS-GFP plus UAS-Tak1 RNAi (bottom panel) in the posterior compartment by the *en*-Gal4 driver stained for lacZ expression (red). Note the increased expression of *ban-lacZ* upon Tak1 knockdown. Scale bars, 50 μm. Quantification of *ban-lacZ* signal is shown in (**M**). Data were analyzed using two-tailed Student's $t$-test and presented as mean ± s.d., $n = 9$ wing discs. RLTV, relative; P, posterior; A, anterior. Data shown are representative of at least three independent experiments. Source data are provided as a Source Data file.

Tak1 is able to activate Hippo signaling by inducing the phosphorylation of Hippo pathway components.

## STRIPAK PP2A complex negatively regulates Tao-1 phosphorylation

Above data indicate that Tak1 functions upstream of Tao-1 to induce the phosphorylation of downstream components of the Hippo pathway. To elucidate the underlying mechanism by which Tak1 promotes Tao-1 phosphorylation, we first performed in vitro kinase assay to examine whether Tak1 can directly phosphorylate bacterially purified GST-Tao-1 fusion protein. In this assay, ATP- γS was used as phosphate donor to generate thiophosphorylated substrate, which further reacts with p-nitrobenzyl mesylate (PNBM) to form a thiophosphate ester that can be detected by thiophosphate-ester-specific antibody[39]. As shown in Supplementary Fig. 2A, we failed to detect phosphorylation of GST-Tao-1 by Tak1 although a known Tak1 substrate, Imd[36], can be strongly phosphorylated by Tak1 in this assay. Thus, it is unlikely that Tak1 functions as a direct Tao-1 kinase.

After excluding Tak1 as a direct Tao-1 kinase, we explored whether Tak1 induces Tao-1 phosphorylation indirectly by regulating the upstream regulators of Tao-1. So far, the regulators of Tao kinase have been barely reported except that in mammalian neurons, the TAO kinase is shown to be regulated by the Mammalian Sterile 20 (Ste20)-like kinase 3 (MST3), a member of the Germinal Center Kinase (GCK) family[40]. Given that many GCK family members are components of the STRIPAK complex[41] and this complex has been shown to dephosphorylate the Ste20 family kinase Hpo[21,22], we explored whether the activity of Tao-1, another Ste20 family kinase, is also regulated by the STRIPAK complex in *Drosophila*. For this purpose, we depleted several essential components of the STRIPAK complex and monitored the phosphorylation of Tao-1 in S2R+ cells. As shown in Fig. 2A, knockdown of major STRIPAK components including Slmap, Fgop2, Cka, Mob4, Strip, Mts and Pp2A-29B all resulted in increased Tao-1 phosphorylation albeit to varying extent, suggesting that STRIPAK complex is a negative regulator of Tao-1. The STRIPAK PP2A phosphatase complex associates with its substrates through different subunits. For example, the Slamp subunit mediates the association between STRIPAK complex and Hpo[22]. Next, we performed co-immunoprecipitation (co-IP) to map the interaction between Tao-1 and STRIPAK subunits. Among all STRIPAK subunits, Slmap, Fgop2 and Cka were detected to physically interact with Tao-1 (Fig. 2B–D and Supplementary Fig. 2B). Interestingly, further analysis indicated that the interaction between Slmap and Tao-1 was reduced by Cka or Fgop2 knockdown (Fig. 2E); the interaction between Fgop2 and Tao-1 was reduced by Cka knockdown, but not Slmap knockdown (Fig. 2F); the interaction between Cka and Tao-1 was not altered by either Slmap or Fgop2 knockdown (Fig. 2G). Collectively, we reason that Cka may function as a bridge between Tao-1 and Slmap or Fgop2 and, by inference, the rest of the STRIPAK complex. To further support this notion, we performed in vitro pull-down assay using bacterially purified GST-Cka, GST-Slmap, GST-Fgop2 and lysates of S2R+ cells transfected with HA-tagged Tao-1. As shown in Fig. 2H–J, association between Tao-1 and Cka, but not Slamp or Fgop2, was detected by this assay. Taken together, these data suggest that the STRIPAK PP2A complex associates with Tao-1 through its subunit Cka and functions as a negative regulator of Tao-1 phosphorylation.

## Tak1 activates Tao-1 by inducing the degradation of Cka through lysosomal pathway

To further elucidate the underlying mechanisms by which Tak1 induces Tao-1 phosphorylation, we tested whether Tak1 was able to disrupt Tao-1-STRIPAK association given that both Tak1 and Tao-1 have been found to physically associate with the STRIPAK complex. Of note, the interaction between Tao-1 and Cka was unaltered by either Tak1 or Tak1$^{S176A}$ (Fig. 3A). However, we noticed that co-transfection of Tak1, but not Tak1$^{S176A}$, greatly reduced the protein level of Cka (Fig. 3A and Supplementary Fig. 3A). Next, we investigated how Tak1 diminishes the protein level of Cka. To this end, we first determined whether Tak1 inhibits the transcription of *Cka*. We overexpressed Tak1 in S2R+ cells and performed quantitative real-time PCR (qRT-PCR) to examine the mRNA level of *Cka*. As shown in Supplementary Fig. 3B, overexpression of Tak1 did not affect *Cka* mRNA level, although the known Tak1 target *Diptericin* (*Dpt*) was dramatically induced by Tak1 overexpression. This result suggested that Tak1 does not inhibit *Cka* transcription.

After excluding a role of Tak1 in suppressing *Cka* transcription, we tested whether Tak1 decreases the protein level of Cka by promoting its degradation. We treated S2R+ cells with translational inhibitor cycloheximide (CHX) and monitored the degradation kinetics of Cka in the absence or presence of Tak1. Notably, the degradation of Cka was dramatically accelerated by Tak1 (Fig. 3B), suggesting that Tak1 indeed promotes the degradation of Cka. Tak1 is an upstream activator of multiple transcription factors, such as Rel, Jun-related antigen (Jra) or Kayak (Kay). The observation that Tak1 is still capable of inducing Cka degradation after CHX treatment strongly suggests that the target gene expression mediated by these transcription factors downstream of Tak1 is unlikely involved in this process. In the cell, the proteasome and lysosome represent two major proteolytic machineries, responsible for the protein degradation in the ubiquitin-proteasome system (UPS) and autophagy, respectively. To define how Tak1 promotes Cka degradation, we examined whether the inhibitors of proteasomal or lysosomal pathways would inhibit the degradation of Cka induced by Tak1. As shown in Fig. 3C, the lysosomal pathway inhibitor Bafilomycin A1 (BafA1) inhibited Tak1-induced Cka degradation. However, the proteasomal inhibitor PS-341 had no effect on the Cka degradation caused by Tak1 (Supplementary Fig. 3C). These results indicate that Tak1 induces the degradation of Cka through the lysosomal pathway rather than the proteasomal pathway. In line with this conclusion, Tak1, but not Tak1$^{S176A}$, can dramatically promote the colocalization between Cka and the lysosomal marker Lamp1 (Supplementary Fig. 3D–G),

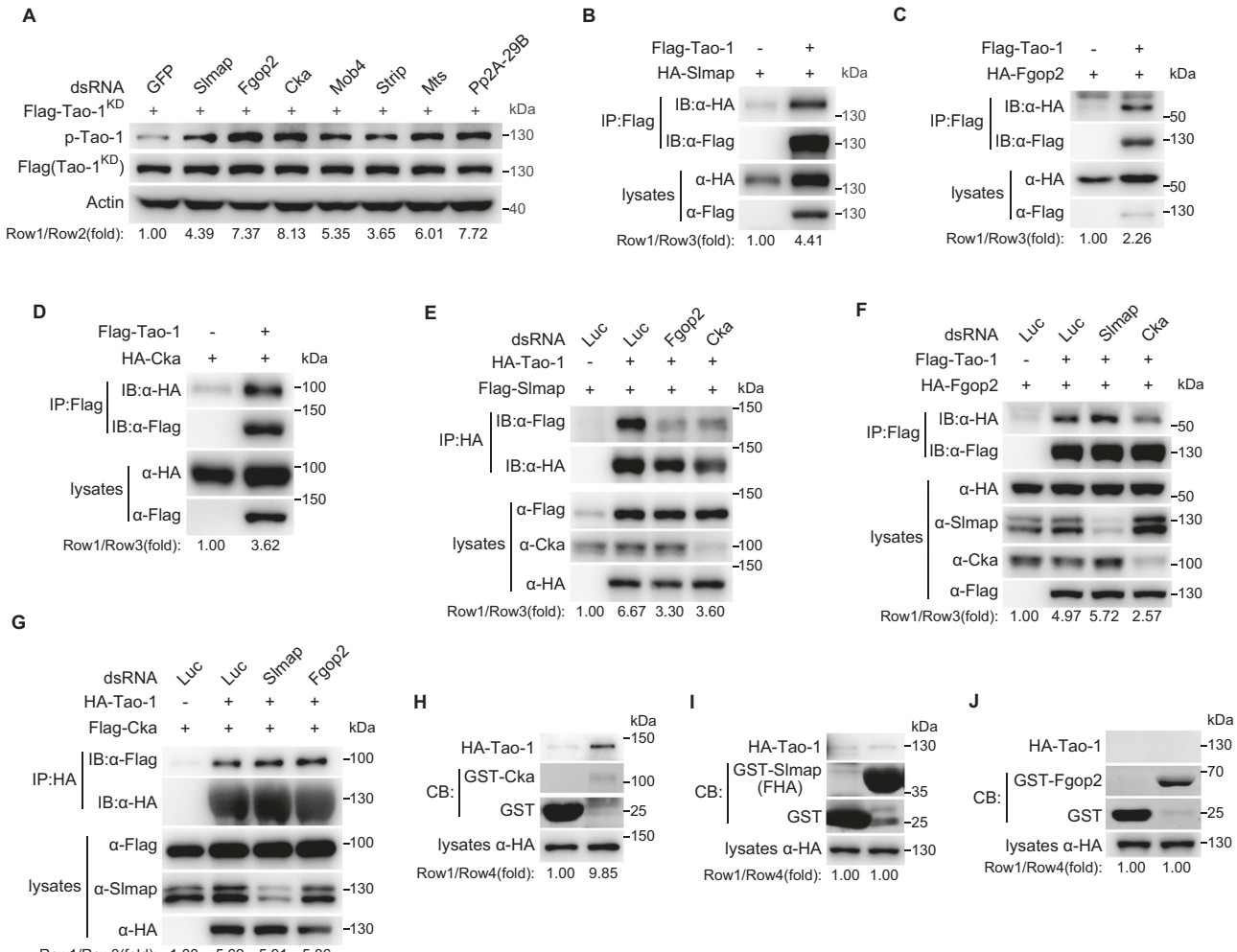

**Fig. 2 | The phosphorylation of Tao-1 is negatively regulated by STRIPAK PP2A complex. A** S2R+ cells were pretreated with dsRNAs of indicated STRIPAK PP2A complex components before transfected with a kinase-dead form of Tao-1 (Flag-Tao-1[KD]). Note the enhanced phosphorylation of Tao-1 upon knockdown of the STRIPAK PP2A complex components. **B–D** Immunoprecipitation was performed in S2R+ cells transfected with indicated plasmids. Note the interaction between Tao-1 and Slmap (**B**), Fgop2 (**C**) or Cka (**D**). **E–G** Immunoprecipitation was performed in S2R+ cells pretreated with indicated dsRNAs then transfected with designated plasmids. Note that Slmap-Tao-1 association was reduced upon knockdown of

Fgop2 or Cka (**E**); Fgop2-Tao-1 association was unaltered by Slmap knockdown, while was reduced by Cka knockdown (**F**); Cka-Tao-1 association was unaltered by either Slmap or Fgop2 knockdown (**G**). GST pull-down assay was performed using cell lysates prepared from S2R+ cells transfected with HA-Tao-1 construct and bacterially purified GST-Cka (**H**), GST-Slmap FHA (**I**) or GST-Fgop2 (**J**) fusion proteins. Note that Tao-1 was pulled down by GST-Cka (**H**), but not GST-Slmap FHA (**I**) or GST-Fgop2 (**J**). FHA, forkhead-associated domain; CB, Coomassie blue. Data shown are representative of at least three independent experiments. Source data are provided as a Source Data file.

indicating enhanced lysosomal localization of Cka upon Tak1 co-expression.

Autophagy, a conserved cellular process that is mediated by a serial of autophagy-related (ATG) proteins, delivers cytoplasmic materials to the lysosomes for degradation[42–45]. Atg8 (LC3/GABARAP in mammals), a key autophagy-related protein in autophagosome formation, mediates the lysosomal degradation of cargo molecules by recruiting them into autophagosomes[46]. Atg8/LC3 protein translocates to newly formed autophagosomes and appears as cytoplasmic puncta upon autophagosomes induction[47], which serves as a typical marker for autophagic flux. To further investigate how Tak1 induces the lysosomal degradation of Cka, we tested whether Tak1 can promote the physical interaction between Cka and Atg8a, given that both Cka and Tak1 had previously been identified as binding partners of *Drosophila* Atg8a[48,49]. As shown by the Co-IP result, Tak1, but not Tak1[S176A], can strongly heighten Cka-Atg8a association (Fig. 3D), suggesting that this effect of Tak1 depends on its kinase activity. Consistently, the enhancing effect of Tak1 on Cka-Atg8a association was reduced by Tak1 inhibitor 5Z-7-Oxozeaenol (5Z-7-O) (Supplementary

Fig. 3H). Moreover, we noted that Tak1-Cka association was mitigated (Fig. 3E), while Tak1-Atg8a association was unaffected by 5Z-7-O (Fig. 3F). These findings suggest a model whereby Tak1 constitutively binds Atg8a, acting as a cargo receptor for the selective autophagy of Cka once Tak1 is activated. This model was further substantiated by immunofluorescent assay performed in *Drosophila* S2R+ cells. When epitope-tagged Cka and Atg8a were co-transfected into S2R+ cells, ~18% of Atg8a puncta were Cka-positive (Fig. 3G, J). Of note, this number was promoted to ~60% by further co-transfection of Tak1, but not Tak1[S176A] (Fig. 3H–J). In agree with the model that Tak1 constitutively binds Atg8a, we observed that almost all Atg8a puncta colocalized with Tak1 or Tak1[S176A]. Moreover, we found that Tak1-induced Cka degradation was inhibited by Atg8 knockdown (Fig. 3K), highlighting the importance of Atg8 in Tak1-mediated lysosomal degradation of Cka. To further validate that the autophagy pathway is required for Tak1-mediated Cka degradation, we tested additional autophagy genes including Atg1, Atg2, or Atg9. Similar to Atg8a, Tak1-induced Cka degradation was inhibited by Atg1, Atg2, or Atg9 knockdown as well (Supplementary Fig. 3I). Taken together, these data

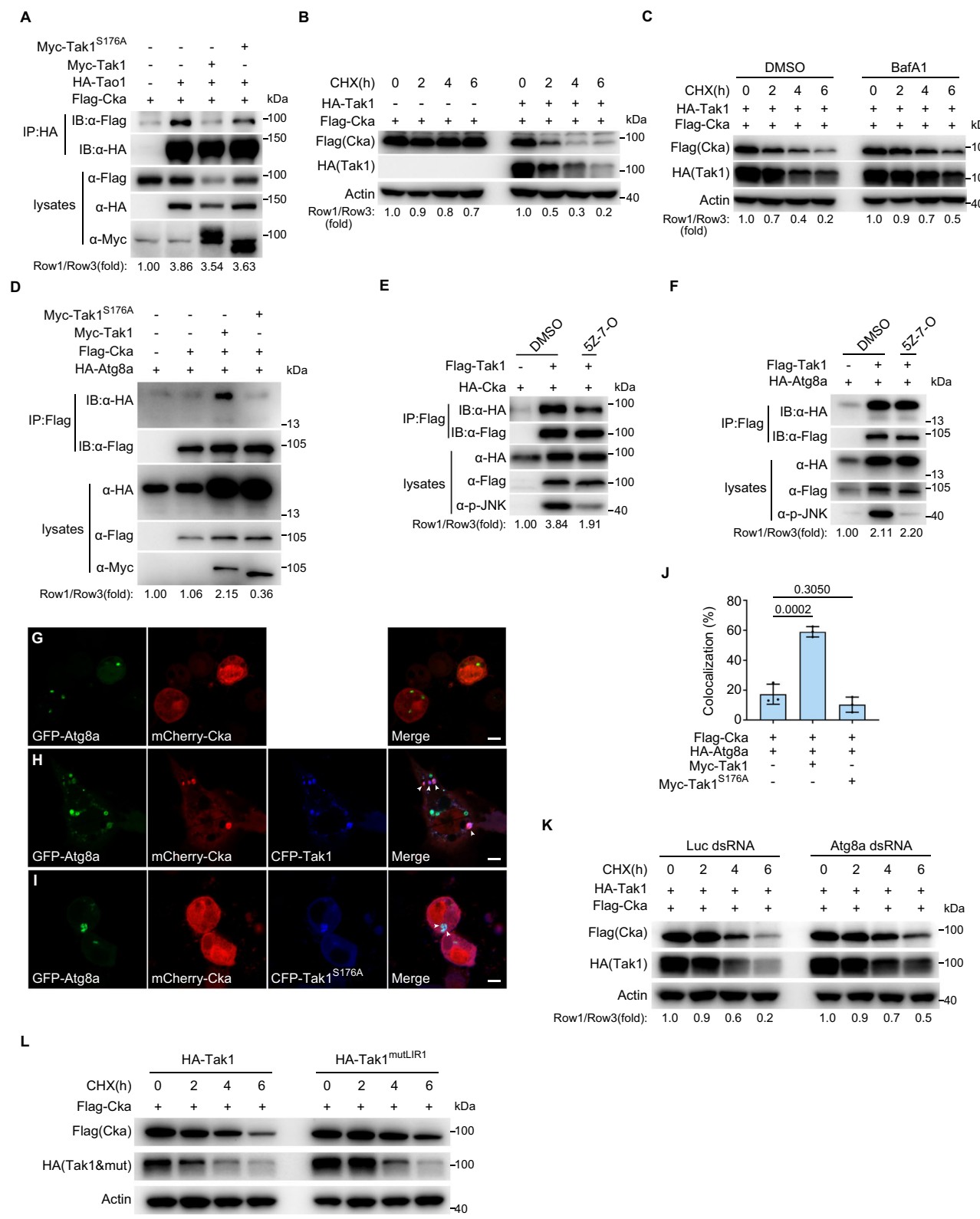

suggest that Tak1 induces the lysosomal degradation of Cka through autophagy pathway likely by promoting Cka-Atg8a association.

Atg8/LC3 interacts with their partners usually through the LC3-interacting regions (LIR) motif found in its binding partners[50,51]. Three conserved LIRs (LIR1, 2, 3) have been identified in Cka using the LIR prediction software iLIR[52], and the LIR2 (aa. 313–318; ANFEFL) is crucial for Atg8a binding[49]. To explore whether LIR2-mediated Atg8a

association plays a role in the selective degradation of Cka, we disrupted the LIR2 of Cka either by substituting the phenylalanine (F) and leucine (L) with alanine (A) as described[49], or by deleting the entire LIR2 motif. Results indicated that Cka mutants with defective LIR2 had comparable degradation kinetics as wild-type Cka (Supplementary Fig. 3J), suggesting that LIR2-mediated Atg8a binding is not essential for Cka degradation. On the other hand, two LIRs have been identified

**Fig. 3 | Tak1 induces the degradation of Cka through lysosomal pathway.**
**A** Immunoprecipitation was performed in S2R+ cells transfected with indicated plasmids. Note the decreased Cka protein level upon co-expression with Tak1 (compare lanes 3 and 2 in the top input blots), but not Tak1[S176A] (compare lanes 4 and 2 in the top input blots). **B** S2R+ cells were transfected with indicated plasmids and treated with CHX (50 μg/mL) for designated times. Note the accelerated degradation of Cka upon Tak1 co-expression. **C** Similar to (**B**) except that DMSO or BafA1 (25 nM) was added together with CHX (50 μg/mL) for indicated times. Note that the accelerated degradation of Cka upon Tak1 co-expression was impeded by BafA1. **D** Immunoprecipitation was performed in S2R+ cells transfected with indicated plasmids. Note that the interaction between Cka and Atg8a was promoted by Tak1, but not Tak1[S176A]. **E, F** Immunoprecipitation was performed in S2R+ cells transfected with indicated plasmids and treated with DMSO or 5Z-7-Oxozeaenol (2 μM, 3 h). Efficient inhibition of Tak1 by 5Z-7-Oxozeaenol was indicated by reduced p-JNK signals (bottom input blots). Note the weakened Tak1-Cka association, while unaltered Tak1-Atg8a association upon 5Z-7-Oxozeaenol treatment. **G–J** Immunostaining performed in live S2R+ cells showing enhanced colocalization of Cka (red) and Atg8a (green) upon co-expression of Tak1, but not Tak1[S176A] (blue) (**G–I**). Arrow heads point to representative puncta with protein colocalization. Quantification of Cka and Atg8a colocalization is shown in (**J**). Data in (**J**) were analyzed using one-way ANOVA, Tukey's HSD test, and presented as mean ± s.d., $n = 3$ biological replicates. Scale bars, 4 μm. **K** S2R+ cells pretreated with dsRNAs of Luciferase or Atg8a and then transfected with indicated plasmids were treated with CHX (50 μg/mL) for indicated times. Note that the accelerated degradation of Cka upon Tak1 co-expression was impeded by Atg8a knockdown. **L** S2R+ cells were transfected with indicated plasmids and treated with CHX (50 μg/mL) for designated times. Note that the degradation of Cka by co-transfection of Tak1[mutLIR1] was slower than that of Tak1. Data shown are representative of at least three independent experiments. Source data are provided as a Source Data file.

in Tak1 which mediate Tak1-Atg8a interaction and the selective degradation of Tak1[48]. Since our model argues that Tak1 acts as a cargo receptor for the selective degradation of Cka, we would expect Tak1-mediated Cka degradation to be dependent on Tak1's LIRs. To test this prediction, we generated Tak1 mutant with inactivated LIR1, the predominant LIR that mediates Tak1-Atg8 association, as previously described[48], and examined its effect on Cka degradation. As shown in Fig. 3L, inactivating LIR1 indeed impaired Tak1's ability of mediating Cka degradation, which further corroborated our model.

## LPS-activated Tak1 induces the lysosomal degradation of endogenous Cka in both S2 cells and fly hemocytes

The results presented so far delineate a model whereby Tak1 induces Cka degradation through lysosomal pathway, thus suppressing the STRIPAK complex. Our data suggest that STRIPAK complex suppression leads to the activation of Tao-1 which then phosphorylates and activates Hpo. Meanwhile, given that STRIPKA complex also functions as a Hpo phosphatase, its suppression can also promote Hippo signaling activation by increasing homeostatic Hpo activity. Since above results were obtained mainly through overexpression of the exogenous epitope-tagged proteins, we further validated the physiological relevance of this model. For this purpose, we first explored whether physiological cues that are known to activate Tak1 can promote the lysosomal degradation of endogenous Cka. We therefore performed CHX pulse chase analysis to monitor the degradation kinetics of endogenous Cka with or without LPS treatment. Notably, LPS treatment dramatically accelerated the degradation of Cka (Fig. 4A), which was inhibited by the inhibitor of lysosomal degradation pathway, $NH_4Cl$ (Fig. 4B), but was not affected by the proteasomal inhibitor PS-341 (Supplementary Fig. 4). LPS-induced Cka degradation was largely suppressed by depletion of Tak1 (Fig. 4C), suggesting that this process is indeed Tak1-dependent. Next, we examined whether LPS treatment can promote Cka-Atg8a association. Due to the lack of Cka and Atg8a antibodies suitable for co-IP, we transiently transfected S2R+ cells with epitope-tagged Cka and Atg8a. Echoing the enhanced Cka-Atg8a association by Tak1 co-transfection, the interaction between Cka and Atg8a was boosted by LPS treatment as well (Fig. 4D). Moreover, dramatically increased number of Cka puncta were observed upon LPS treatment, and these Cka puncta localized in the lysosomes labeled by Lamp1-GFP (Fig. 4E, F). Taken together, these results indicate that LPS-activated Tak1 induces the lysosomal degradation of endogenous Cka.

Since S2 cells were derived from *Drosophila* embryonic hemocytes[53], we therefore further verified above model in primary hemocytes. Due to the technical constraints of primary hemocyte isolation, we were unable to collect enough samples for CHX pulse chase analysis to monitor the degradation kinetics of endogenous Cka. Instead, we examined the protein level of endogenous Cka upon Tak1 overexpression, or at one fixed time point after LPS treatment. Consistent with the results in S2 cells, we observed reduced level of Cka upon Tak1 overexpression driven by hemocyte-specific *Hml*-Gal4, which can be suppressed by simultaneous knockdown of multiple autophagy genes such as Atg1, Atg9, Atg13 or Atg101 in *Drosophila* (Fig. 4G). Moreover, LPS-treatment resulted in reduction of Cka level in primary hemocytes of wild-type flies (Fig. 4H). Echoing the reduced protein level of Cka, qRT-PCR result indicated that LPS treatment resulted in decreased expression of Yki target gene *expanded* (*ex*) (Fig. 4I), suggesting Hippo signaling activation upon LPS treatment in primary hemocytes. However, LPS failed to reduce the protein level of Cka in primary hemocytes isolated from *Tak1* knockout flies (Fig. 4H), suggesting the reduction of Cka level by LPS in hemocytes is Tak1-dependent. Interestingly, we noted that the level of endogenous Cka is lower in Tak1 mutant hemocytes than that in wild-type hemocytes. This is likely because the homeostatic protein levels of endogenous Cka is also controlled by other mechanisms besides Tak1, such as Pelle[30].

## Tak1-mediated Hippo signaling regulates the phagocytic activity of hemocytes

Next, we further explore the role of Tak1-mediated Hippo signaling in hemocyte physiology. There are three subtypes of hemocytes in *Drosophila*, namely plasmatocytes, crystal cells and lamellocytes[24,54]. The plasmatocyte is the most abundant cell type comprising ~95% of the hemocyte repertoire, and is macrophage-like phagocyte. The crystal cells represent ~5% of hemocytes in circulation, and are involved in the melanization of invading organisms by secreting components of the phenol oxidase cascade. The lamellocytes are rarely observed in healthy animal but can be induced to differentiate specifically by parasitic wasp infestation[55]. Given that the plasmatocytes are the predominant hemocytes and their major function is to remove invading pathogens via phagocytosis, to gain further insight into the role of Tak1-mediated Hippo signaling in hemocyte physiology, we examined the phagocytic ability of the hemocytes. For this purpose, primary hemocytes were isolated from 3rd instar larvae and incubated with GFP-positive *E. coli* (GFP-*E. coli*) in vitro. After quenching the extracellular GFP-*E. coli* with trypan blue, the percentage of cells with intracellular GFP-*E. coli* was determined via flow cytometry. Supplementary Fig. 5A shows the gating strategies for this assay. Notably, depletion of Tak1 using the hemocyte-specific *Hml*-Gal4 driver[56] attenuated the phagocytic ability of the hemocytes (Fig. 5A). Overexpression of Yki mimicked Tak1 knockdown phenotype (Fig. 5B) and simultaneous knockdown of Yki fully rescued the reduced phagocytic ability of the hemocytes with Tak1 knockdown (Fig. 5C), indicating the compromised phagocytic ability of Tak1 knockdown was mediated by Yki. On the other hand, overexpression of Tak1 strongly promoted the phagocytic ability of the hemocytes (Fig. 5D) and this phenotype was partially rescued by knockdown of Hippo pathway components such as Tao-1 or Hpo (Fig. 5E), further highlighting the pivotal roles of Tak1-mediated

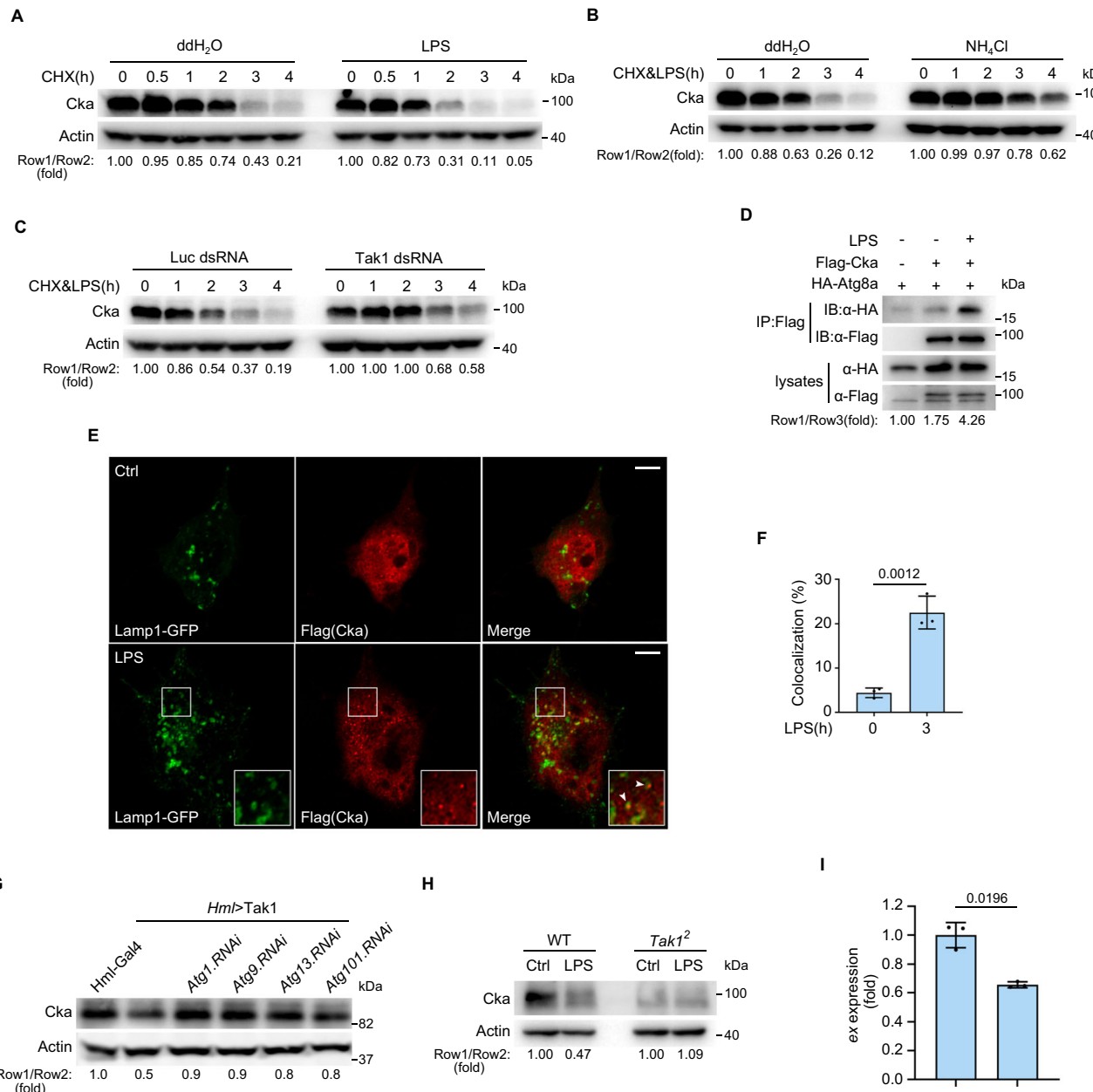

**Fig. 4 | LPS-activated Tak1 induces the lysosomal degradation of endogenous Cka.** In (**A**–**D**), *Drosophila* S2 cells were pretreated with 20-Hydroxyecdysone (1 μM, 48 h) to potentiate the immune response before subsequent treatment. **A** S2 cells were treated with CHX (50 μg/mL) together with ddH₂O or LPS (10 μg/mL) for indicated times. Note the accelerated degradation of endogenous Cka upon LPS treatment. **B** Similar to (**A**) except that ddH₂O or NH₄Cl (40 mM) was added together with CHX (50 μg/mL) and LPS (10 μg/mL) for indicated times. Note that the accelerated degradation of endogenous Cka upon LPS treatment was hampered by NH₄Cl. **C** S2 cells pretreated with dsRNAs of Luciferase or Tak1 were treated with CHX (50 μg/mL) and LPS (10 μg/mL) for indicated times. **D** Immunoprecipitation was performed in S2 cells transfected with indicated plasmids and treated with ddH₂O or LPS (10 μg/mL, 2 h). Note the enhanced interaction between Cka and Atg8a upon LPS treatment. **E**, **F** Immunostaining was performed in S2R+ cells transfected with Flag-Cka and Lamp1-GFP constructs,

showing enhanced colocalization of Cka (red) and Lamp1 (green) upon LPS treatment (10 μg/mL, 3 h). Arrow heads are representative Cka puncta colocalized with Lamp1-GFP. Quantification of Cka and Atg8a colocalization is shown in (**F**). Data were analyzed using two-tailed Student's *t*-test and presented as mean ± s.d., *n* = 3 biological replicates. Scale bars, 5 μm. **G** Endogenous Cka level was examined in stage 17 embryos with indicated genotype via western blot. **H** Primary hemocytes isolated from wild-type or *Tak1* null (*Tak1²*) flies were treated with LPS (10 μg/mL, 0.5 h). Note the decreased endogenous Cka level in wild-type hemocytes and unaltered endogenous Cka level in *Tak1²* hemocytes upon LPS treatment. **I** qRT-PCR assay performed in primary hemocytes showing that mRNA level of *ex* was reduced upon LPS treatment (10 μg/mL, 0.5 h). Data were analyzed using two-tailed Student's *t* test and presented as mean ± s.d., *n* = 3 biological replicates. Data shown are representative of at least three independent experiments. Source data are provided as a Source Data file.

Hippo signaling in maintaining the phagocytic ability of the hemocytes. To further refine these findings, we determined whether the observed differential phagocytic ability was attributed to changes of the identity of the hemocyte since the Hippo signaling has been implicated in fly hemocyte differentiation[57,58]. To this end, we

performed Giemsa staining of hemolymph smears[59] and counted differential hemocytes. The result indicated that the proportion of the plasmatocytes was comparable among above flies (Supplementary Fig. 5B). This result ruled out the possibility that the differential phagocytic ability was due to the change of the hemocyte identity.

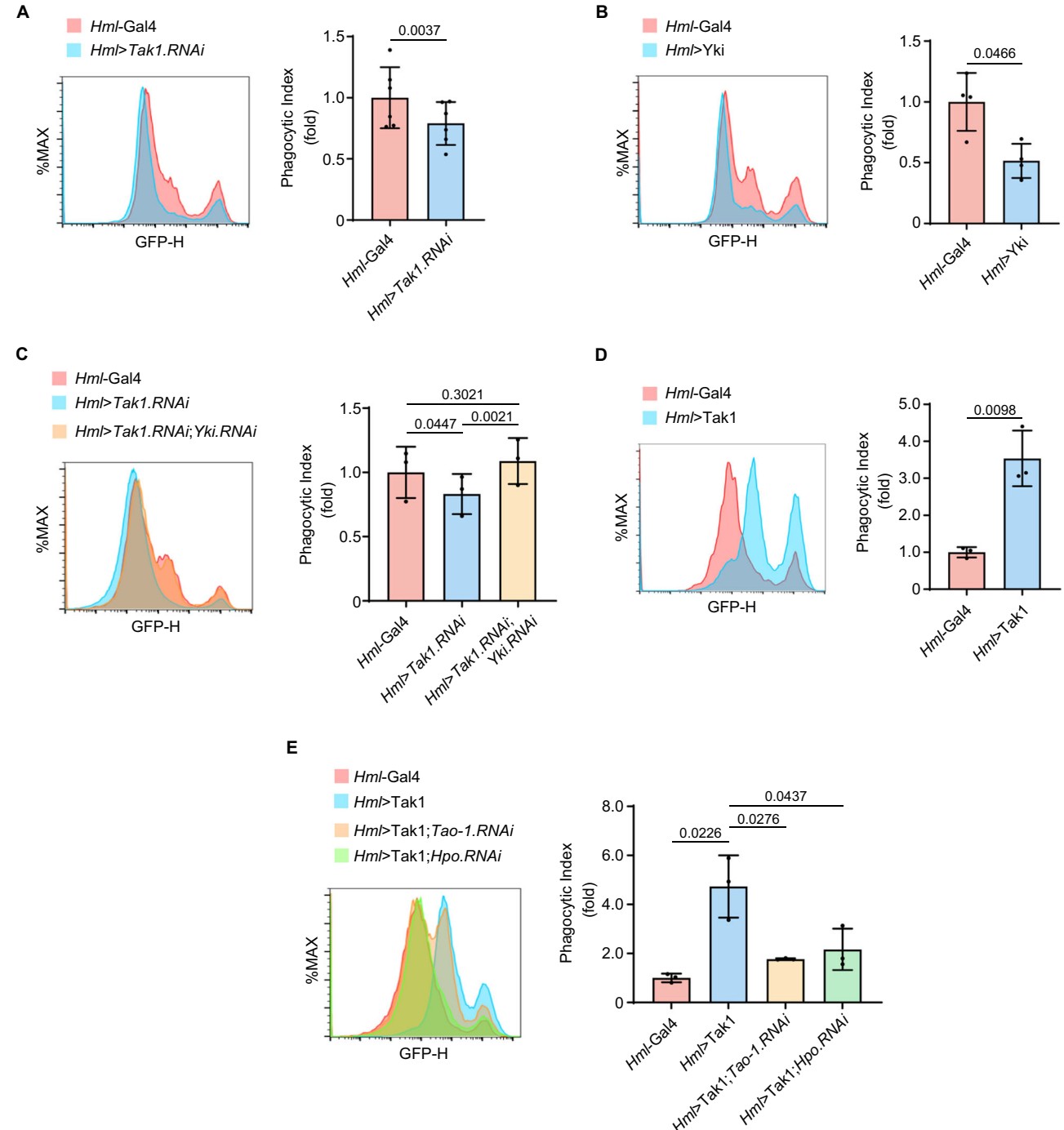

**Fig. 5 | Tak1-mediated Hippo signaling regulates the phagocytic activity of hemocytes.** Flow cytometry analysis was performed in the primary hemocytes isolated from 3rd instar larvae of indicated genotypes. **A** Hemocytes with Tak1 depletion showed significantly reduced phagocytic activity. **B** Hemocytes with Yki overexpression showed significantly reduced phagocytic activity. **C** The reduced phagocytic activity of Tak1 depleting hemocytes was fully rescued by simultaneous knockdown of Yki. **D** Hemocytes with Tak1 overexpression showed significantly increased phagocytic activity. **E** The enhanced phagocytic activity of Tak1 overexpression hemocytes was rescued by knockdown of Tao-1 or Hpo. Left panel in each figure is representative Flow cytometry histogram of three independent experiments analyzed using FlowJo software. Histogram overlays are displayed as % Max, scaling each curve to mode = 100%. Data were analyzed using two-tailed Student's *t* test and presented as mean ± s.d., (**A**: $n = 6$ biological replicates; **B**: $n = 4$ biological replicates; **C**: $n = 3$ biological replicates; **D**: $n = 3$ biological replicates; **E**: $n = 3$ biological replicates). Source data are provided as a Source Data file.

## Tak1-mediated Hippo signaling controls immunological memory in *Drosophila*

Accumulating evidence indicates that innate immunity, like adaptive immunity, also shows memory traits[60,61]. Given that the phagocytic ability of the hemocyte is crucial for innate immune memory in *Drosophila*[62,63], we further explored whether Tak1-mediated Hippo signaling controls fly innate immunity memory. To this end, we exploited a bacterial combination which was recently-reported to elicit immune memory in *Drosophila*[64]. In this combination, the low pathogenic *Micrococcus luteus* (*M. luteus*) was used for training and the highly pathogenic *Staphylococcus aureus* (*S. aureus*) was used for subsequent challenge. We first further characterized this bacterial

combination using *Drosophila* S2 cells. The expression of multiple AMP such as *Dpt* or *Metchnikowin* (*Mtk*) were rapidly induced by *M. luteus* treatment and gradually returned to basal level after *M. luteus* removal (Supplementary Fig. 6A–C). Strikingly, subsequent *S. aureus* treatment provoked significantly higher *Dpt* or *Mtk* expression levels in cells with *M. luteus* pretreatment than cells without *M. luteus* pretreatment (Supplementary Fig. 6A–C), indicating heightened innate immune responses, a major characteristic of trained innate immunity[60]. Consistent with the innate immune memory elicited by this bacterial combination, *Hml*-Gal4 control flies that were pre-infected with *M. luteus* showed higher survival rate than those that were preinjected with PBS upon *S. aureus* infection (Fig. 6A, B). Notably, Tak1 depletion in hemocytes significantly compromised the innate immune memory of the fly (Fig. 6A), while Tak1 overexpression in hemocytes dramatically potentiated the fly innate immune memory (Fig. 6B). Strikingly, neither Tak1 knockdown nor Tak1 overexpression in hemocytes affected the survival rate of PBS-preinjected flies upon *S. aureus* infection (Fig. 6A, B), suggesting that Tak1 does not control cellular immune reactions per se but only has an impact on the innate immune memory of the fly. Further analysis indicated that simultaneous depletion of Yki rescued the compromised innate immune memory of flies with Tak1 knockdown in hemocytes (Fig. 6C). On the other hand, the enhanced innate immune memory in flies with hemocyte overexpression of Tak1 was rescued by concomitant depletion of Tao-1 (Fig. 6D) or Hpo (Fig. 6E). In addition, we determined that the densities of the hemocytes were comparable among these flies (Fig. 6F), which ruled out the possibility that the varied innate immune memory phenotype was due to the change of the hemocyte number. Taken together, these data indicated that Tak1-mediated Hippo signaling controls innate immune memory in *Drosophila*.

### Hippo signaling is involved in Tak1-induced cell death

After establishing a role for Tak1-mediated Hippo signaling in the immunological memory of *Drosophila*, we wished to examine the requirement of this regulation in additional physiological settings. Ectopic Tak1 activity induces apoptosis by activating the JNK pathway and results in tissue ablation in *Drosophila*[37]. Given that our results identified Tak1 as a Hippo-activating kinase and activation of the Hippo signaling is known to cause apoptosis, we postulate that the Hippo signaling is involved in Tak1-induced apoptosis and tissue ablation. To test this hypothesis, we ectopically expressed Tak1 in fly eyes using eye-specific *sev*-Gal4 driver and examined whether manipulating Hippo signaling components would rescue Tak1-induced apoptosis and tissue ablation. As the previous report, we observed reduced size of adult eyes with Tak1 overexpression (Supplementary Fig. 7A, B). Notably, this Tak1-induced small eye phenotype was dramatically rescued by simultaneous depletion of Tao-1 or Wts (Supplementary Fig. 7A, B), suggesting the involvement of Hippo signaling in Tak1-induced tissue ablation. Next, we stained the larval eye imaginal discs for cleaved Dcp-1, an effector caspase in *Drosophila*, to monitor apoptotic cell death. Consistent with the adult eye size phenotype, Tak1 overexpression promoted apoptosis, which was largely inhibited by concurrent Tao-1 or Wts knockdown (Supplementary Fig. 7C, D). Since our data indicated that the autophagy pathway is required for Tak1-mediated Cka degradation and Hippo pathway activation, we predicted that interfering with the autophagy pathway would inhibit the tissue ablation and cell death resulted from Tak1 expression. Indeed, depletions of multiple *Atg* genes suppressed Tak1-induced small eye phenotype (Supplementary Fig. 7A, B) and apoptotic cell death (Supplementary Fig. 7C, D).

### The regulation of the Hippo signaling by Tak1 is evolutionarily conserved

Having demonstrated the regulation of the Hippo signaling by Tak1 in *Drosophila*, we wished to examine whether a similar mechanism operates in mammalian cells. To this end, we co-transfected epitope-tagged MAP3K7 (mammalian Tak1) and components of the mammalian Hippo pathway into HEK293T cells. Similar to their *Drosophila* counterparts, the phosphorylation of TAOK1, MST1/2, LATS1/2 or YAP was all induced by MAP3K7 (Fig. 7A–F). MAP3K7-induced YAP S127 phosphorylation was blocked in *LATS1/2* null cells (Fig. 7G), indicating that this process is LATS1/2-dependent. It is known that YAP S127 phosphorylation results in the cytoplasmic retention and inactivation of YAP. Consistent with this notion, MAP3K7 overexpression in HEK293T cells reduced the mRNA levels of YAP target genes such as *CTGF*, *CYR61* or *AJUBA* (Supplementary Fig. 8A). On the other hand, the mRNA levels of these YAP targets genes were increased in *MAP3K7* knockout cell lines (Supplementary Fig. 8B). Akin to their *Drosophila* homologs, MAP3K7 induced the degradation of STRN1 (mammalian Cka) (Fig. 7H) and this process was inhibited by BafA1 (Fig. 7I), but not PS-341 (Supplementary Fig. 9A). Moreover, inhibition of the autophagy using ULK-101, a potent inhibitor of ULK1/2 (mammalian Atg1), suppressed MAP3K7-induced STRN1 degradation (Supplementary Fig. 9B). These results indicate that MAP3K7 plays a conserved role in regulating the Hippo pathway activity.

To further attest the conservation and validate the physiological relevance in mammalian cells, we treated human THP-1 cells with the bacterial component LPS. Similar to the results in *Drosophila*, LPS treatment induced degradation of endogenous STRN1 and phosphorylation of the Hippo pathway components LATS1 and MOB1 in human THP-1 cells (Fig. 7J). Moreover, LPS treatment dramatically promoted the association between STRN1 and LC3B (Fig. 7K). Like the immune stimuli LPS, the proinflammatory cytokine TNFα also induced the degradation of STRN1 and phosphorylation of the Hippo pathway components in MEF cells (Fig. 7L). In addition, TNFα treatment strongly heightened the association between STRN1 and LC3B in WT but not *MAP3K7* knockout HEK293T cells (Fig. 7M), indicating the dependency on *MAP3K7* in this process.

## Discussion

Although the Hippo signaling pathway was initially defined as an intracellular signaling cascade controlling tissue growth, differentiation and regeneration, accumulating evidence indicates that it participates in non-growth-related processes such as innate immunity[65–67]. The Hippo signaling can be triggered by immune or proinflammatory stimuli, yet the underlying mechanism remains largely unclear. In this study, we have elucidated a Tak1-STRIPAK-Tao axis that mediates Hippo pathway activation by innate immune and proinflammatory stimuli (Fig. 7N), and uncovered an essential role for Hippo signaling in controlling the phagocytic ability of hemocytes and innate immune memory. These findings not only extended our understanding of the physiological functions of this important signaling cascade, but also suggest the Hippo signaling pathway as an important effector of innate immune and proinflammatory signaling.

Although our study reveals an essential role for Tak1-mediated Hippo signaling in regulating the phagocytosis of *Drosophila* hemocytes, the underlying mechanisms remain undecided. The phagocytosis process involves two key steps: (1) phagocytic receptors-mediated binding of the target to the phagocyte, (2) target internalization through plasma membrane remodeling[68,69]. A deficiency in either of these two steps would result in compromised phagocytosis. Interestingly, the Hippo signaling has been widely implicated in regulating remodeling of actin cytoskeleton and plasma membrane[70]. Therefore, it would be intriguing to examine whether Tak1-mediated Hippo signaling modulates phagocytosis by controlling actin/plasma membrane remodeling. Moreover, given the importance of phagocytic receptors in tethering the target, whether Tak1-mediated Hippo signaling can control the expression of the phagocytic receptors is another appealing question. Besides in immune response, phagocytosis by hemocyte also plays important roles in extra processes such as

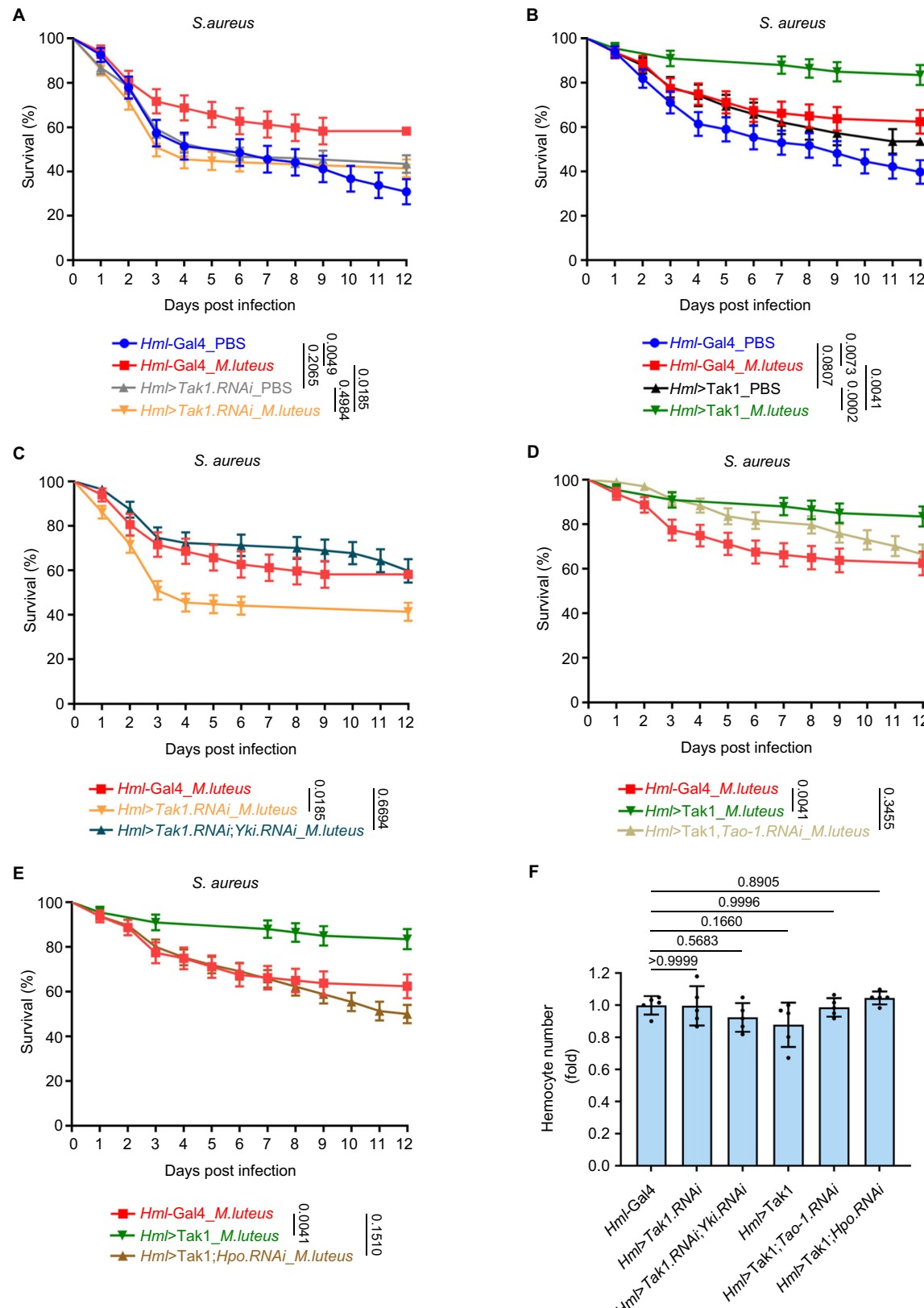

development and tissue homeostasis through removal of apoptotic/damaged cells[71,72]. Thus, whether disruption of the Hippo signaling in hemocytes would result in developmental or regeneration defects warrants further investigation.

Previous study shows that the Hippo pathway is involved in the humoral branch of fly innate immunity through cross talking with the Toll pathway in fly fat body, a major tissue of humoral immune response[30]. Meanwhile, the same study indicates that the Toll signaling can also regulate the Hippo signaling in S2 cells, a cell line which has similar phagocytic ability as the primary hemocyte[73], suggesting that the Toll-Hippo signaling axis also operates in fly hemocytes, although the role of Toll-Hippo axis in hemocyte physiology remains

**Fig. 6 | Tak1-mediated Hippo signaling controls innate immune memory in *Drosophila*. A** Control flies exhibited innate immune memory character which was compromised by Tak1 depletion. Survival curves were analyzed using log-rank (Mantel Cox) test and presented as mean ± s.e.m., (*Hml-Gal4*_PBS: *n* = 68 flies from three independent experiments; *Hml-Gal4_M. luteus*: *n* = 67 flies from three independent experiments; *Hml>Tak1.RNAi*_PBS: *n* = 152 flies from three independent experiments; *Hml>Tak1.RNAi_M. luteus*: *n* = 145 flies from three independent experiments). **B** Tak1 overexpression in hemocytes heightened innate immune memory. Survival curves were analyzed using log-rank (Mantel Cox) test and presented as mean ± s.e.m., (*Hml-Gal4*_PBS: *n* = 83 flies from three independent experiments; *Hml-Gal4_M. luteus*: *n* = 80 flies from three independent experiments; *Hml>Tak1*_PBS: *n* = 82 flies from three independent experiments; *Hml>Tak1_M. luteus*: *n* = 67 flies from three independent experiments). **C** The compromised innate immune memory in Tak1 knockdown flies was rescued by Yki knockdown. Survival curves were analyzed using log-rank (Mantel Cox) test and presented as mean ± s.e.m., (*Hml-Gal4_M. luteus*: *n* = 67 flies from three independent experiments; *Hml>Tak1.RNAi_M. luteus*: *n* = 145 flies from three

independent experiments; *Hml>Tak1.RNAi;Yki.RNAi_M. luteus*: *n* = 87 flies from three independent experiments). **D** The immune memory in Tak1 overexpression flies was rescued by concomitant depletion of Tao-1. Survival curves were analyzed using log-rank (Mantel Cox) test and presented as mean ± s.e.m., (*Hml-Gal4_M. luteus*: *n* = 80 flies from three independent experiments; *Hml>Tak1_M. luteus*: *n* = 67 flies from three independent experiments; *Hml>Tak1;Tao-1.RNAi_M. luteus*: *n* = 104 flies from three independent experiments). **E** The immune memory in Tak1 overexpression flies was rescued by concomitant depletion of Hpo. Survival curves were analyzed using log-rank (Mantel Cox) test and presented as mean ± s.e.m., (*Hml-Gal4_M. luteus*: *n* = 80 flies from three independent experiments; *Hml>Tak1_M. luteus*: *n* = 67 flies from three independent experiments; *Hml>Tak1;Hpo.RNAi_M. luteus*: *n* = 146 flies from three independent experiments). **F** The relative hemocyte number of the flies was determined via a hemocytometer. Data were analyzed using one-way ANOVA with Tukey's HSD test and presented as mean ± s.d., *n* = 5 biological replicates. Source data are provided as a Source Data file.

---

unknown. Interestingly, our current study identifies the Imd pathway component Tak1 as a Hippo-activating kinase in fly hemocytes, and reveals that the Hippo pathway plays a role in hemocyte phagocytosis and immunological memory of *Drosophila*. Thus, the Hippo pathway plays a broad role in fly innate immune response. Since both the Toll and the Imd signaling can activate the Hippo signaling in the hemocyte, it would be intriguing to tell the relative contribution of the two branches to the Hippo signaling, and to examine whether the Toll and the Imd signaling have synergistic effect on Hippo pathway regulation. These questions can be answered through analyzing the Hippo signaling activity in Toll single mutant, Tak1 single mutant, and Toll/Tak1 double mutant hemocytes. Moreover, comparing the transcriptome of Toll or Tak1 mutant hemocytes through RNA sequencing will reveal whether the Toll-Hippo axis and the Tak1-Hippo axis control the same set of target genes, which will shed light on the physiological roles of Toll-Hippo axis in fly hemocyte. These appealing questions warrant further investigation.

Our mechanistic interrogation of Tak1-induced Cka degradation reveals that this process is through the autophagic-lysosomal degradation pathway rather than the ubiquitin-proteasome system. The result that Tak1 promotes the interaction between Cka and Atg8a indicates that Tak1 acts as a cargo receptor for the selective autophagy of Cka. Interestingly, Kenny (Key), a downstream kinase of Tak1 in the Imd pathway, has also been identified as a selective autophagy receptor[74]. Whether Kenny can also regulate the Hippo signaling in a similar manner to Tak1 would be an intriguing question. The connections between Hippo signaling and autophagy have been described by multiple studies[75–77]. Our findings in this study provide additional insight into the crosstalk between autophagy and Hippo signaling, demonstrating activation of Hippo signaling by autophagic-lysosomal degradation of a negative regulator, Cka/STRN.

Accumulating evidence shows that innate immune response also exhibits adaptive characteristics in that priming the host with a sub-lethal dose of a pathogen protects against an otherwise-lethal second challenge of the same or a different pathogen. Our current study indicates that Tak1-mediated Hippo signaling is involved in the innate immune memory in *Drosophila*, yet the underlying mechanisms warrant further investigation. Previous studies have shown that epigenetic and metabolic reprogramming play important roles in immune memory in mammals[60,61]. Given that the Hippo signaling has been widely linked to the control of epigenetic remodeling[78–80] or metabolic reprogramming[81–84], it would be worthwhile to investigate whether the Hippo pathway is involved in the fly innate immune memory by regulating epigenetic and metabolic changes as well. Moreover, our data indicate that the regulation of the Hippo pathway by Tak1 is conserved in mammal, investigating whether the Hippo pathway also participates in the mammalian trained immunity would be intriguing.

## Methods
### Drosophila genetics
Stocks were raised on standard cornmeal-agar medium. Stag 17 embryos, 3rd instar larvae, 3- to 6-days old adults were used in this study. *w1118* flies were used as a standard wild-type strain. *Hml*-Gal4 (stock ID 30141), *Hml* > GFP (stock ID 30142), *en*-Gal4 (stock ID 83350), *sev*-Gal4 (stock ID 5793), UAS-Tak1 (stock ID 58810), *Tak1²* (stock ID 26272), Atg1 RNAi (stock ID 26731), Atg101 RNAi (stock ID 34360) and *ban*-lacZ (stock ID 10154) lines were obtained from Bloomington Drosophila Stock Center (BDSC). Tak1 RNAi (stock ID 101357), Tao-1 RNAi (stock ID 17432), Hpo RNAi (stock ID 104169), Wts RNAi (stock ID 106174), Atg13 RNAi (stock ID 27955) and Yki RNAi (stock ID 40497) lines were collected from Vienna Drosophila Resource Center (VDRC). Atg9 RNAi (stock ID THU2895) was collected from TsingHua Fly Center. UAS-Yki line has been reported previously[85].

### Microbial strains and cell lines
*M. luteus* (ATCC 4698) and *S. aureus* (ATCC 25923) were cultured in tryptic soy broth. GFP-*E. coli* was generated as described[32] and cultured in LB medium. *Drosophila* S2 cells (RRID: CVCL_TZ72) and S2R+ cells (RRID: CVCL_Z831) were cultured at 25 °C in *Drosophila* Schneider's Medium (Biological Industries) supplemented with 10% FBS and antibiotics. Human HEK293T cells (ATCC#CRL-11268) and mouse MEF cells (ATCC#CRL-2991) were cultured at 37 °C in DMEM medium (Sigma) supplemented with 10% FBS and antibiotics. Human THP-1 cells (ATCC#TIB-202) were cultured at 37 °C in RPMI 1640 medium (Sigma) supplemented with 10% FBS and antibiotics.

### Plasmids and antibodies
Flag-Cka, GST-Cka, Flag-Slmap, HA-Slmap, GST-Slmap (FHA), Flag-Fgop2, Flag-Strip, Flag-Mob4, Flag-Mts, Flag-PP2A-29B, Myc-Hpo$^{KD}$, HA-Yki, V5-Wts, Flag-Tao-1, Flag-Tao-1$^{KD}$, Flag-Mats, HA-Tak1, GST-Imd, Flag-YAP, Flag-MST1$^{KD}$, Flag-MST2$^{KD}$, Myc-LATS1$^{KD}$, Myc-LATS2$^{KD}$, Tak1$^{mutLIR1}$, and Cka$^{mutLIR2}$ constructs have been described previously[16,17,19,22,30,48,49,85–87]. All other related constructs of point mutation, truncation or different epitope tag were generated from the constructs described above. GFP-LC3 was kindly gifted by Qiao Wu (Xiamen University). Flag-MAP3K7 was made from a plasmid obtained from Addgene. HA-STRN1 was generated by a cDNA clone collected by School of Life Sciences, Xiamen University. HA-Atg8a was made from a cDNA clone obtained from the Drosophila Genomics Resource Center (DGRC). LAMP-GFP was synthesized by Tsingke Biotechnology. All plasmids were verified by DNA sequencing (Tsingke Biotechnology).

The following primary antibodies were used in this study: β-Actin (CST #4967, 1:10,000), HA (CST #3724, 1:1000), p-MST1/2 T183/T180 (also used to detect p-Hpo, CST #49332, 1:1000), p-LATS1/2 T1079/ T1041 (CST #8654, 1:1000), p-JNK T183/Y185 (CST #4668, 1:1000),

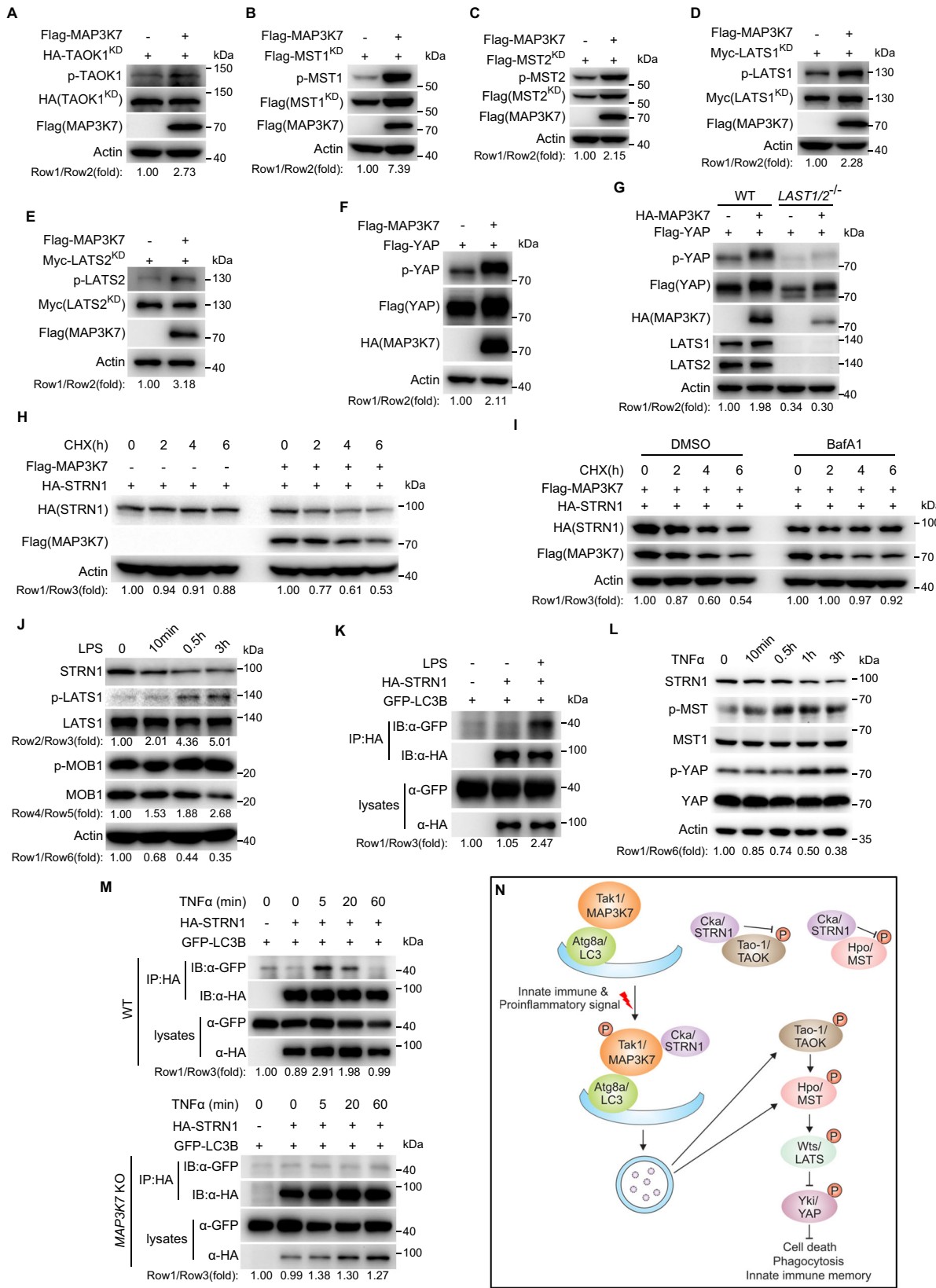

p-YAP S127 (CST #4911, 1:1000), p-MOB T35 (also used to detect p-Mats, CST #3863, 1:1000), LATS1/2 (CST #3477, CST #5888, 1:1000), cleaved Dcp-1 (CST #9578, 1:200) and GFP (CST #2956, 1:1000); p-TAOK2 S181 (also used to detect p-Tao-1; R&D Systems #PPS037, 1:1000); β-galactosidase (Developmental Studies Hybridoma Bank #40-1a, 1:100); STRN1 (Abclonal #A7734, 1:1000); Flag (Sigma #F1804, 1:5000), HA (Sigma #H9658, 1:3000) and c-Myc (Sigma #OP10L, 1:1000); Flag (Biolegend #637304, 1:5000); Thiophosphate ester (Abcam #ab133473, 1:1000); p-Wts T1077, p-Yki S168, Hpo, Mats, Cka, Yki and Slmap (from D. Pan, 1:1000). The following secondary antibodies were used in this study: Goat anti-rat IgG HRP (Jackson ImmunoResearch #112-035-003, 1:5000); Goat anti-rabbit IgG HRP (Jackson

**Fig. 7 | The regulation of the Hippo signaling by Tak1 is evolutionarily conserved. A–F** HEK293T cells were transfected with indicated plasmids. The phosphorylation of TAOK1, MST1/2, LATS1/2 and YAP was detected via p-TAOK2 S181 (**A**), p-MST1/2 T183/T180 (**B**, **C**), p-LATS1/2 T1079/T1041 (**D**, **E**) and p-YAP S127 (**F**) antibodies. The kinase-dead (KD) form of indicated kinases was used to avoid their potential autophosphorylation. **G** Wild-type or *LATS1/2* null HEK293T cells were transfected with indicated plasmids. Note that MAP3K7-induced YAP phosphorylation was inhibited in *LATS1/2* null cells. **H** HEK293T cells transfected with indicated plasmids were treated with CHX (50 μg/mL) for designated times. Note the accelerated degradation of STRN1 upon MAP3K7 co-expression. **I** Similar to (**H**) except that DMSO or BafA1 (25 nM) was added together with CHX (50 μg/mL) for indicated times. Note that the accelerated degradation of STRN1 upon MAP3K7 co-expression was impeded by BafA1. **J** Human THP-1 monocytes were differentiated into macrophages with phorbol 12-myristate 13-acetate (PMA, 100 nM, 24 h) before treated with LPS (10 μg/mL) for indicated times. Note the decreased level of endogenous STRN1 and enhanced phosphorylation of endogenous LATS1 or MOB1 upon LPS treatment. **K** HEK293T cells were transfected with indicated plasmids before treated with LPS (10 μg/mL, 1 h). Immunoprecipitation was performed to observe the interaction of STRN1 and LC3B. **L** MEF cells were treated with TNFα (20 ng/mL) for indicated times. Note the decreased level of endogenous STRN1 and enhanced phosphorylation of endogenous MST1 or YAP upon TNFα treatment. **M** Similar to (**K**) except that TNFα (10 ng/mL) was used to treat the cells. Note the heightened STRN1-LC3B association in WT but not *MAP3K7* knockout cells upon TNFα treatment. **N** A schematic model depicting the mechanism of Tak1-mediated activation of the Hippo signaling by innate immune or proinflammatory signal. Activated Tak1/MAP3K7 promotes Atg8a/LC3-Cka/STRN1 association and mediate the lysosomal degradation of Cka/STRN1, the negative regulator of Tao-1/TAOK and Hpo/MST. Activated Hippo signaling is involved in processes including cell death, phagocytosis and innate immune memory. Data shown are representative of at least three independent experiments. Source data are provided as a Source Data file.

ImmunoResearch #111-035-003, 1:5000); Goat anti-mouse IgG HRP (Jackson ImmunoResearch #115-035-003, 1:5000); Goat anti-rabbit IgG Cy3 (Jackson ImmunoResearch #111-165-003, 1:500); Goat anti-rabbit IgG FITC (Jackson ImmunoResearch #111-095-003, 1:500); Goat anti-mouse IgG Cy3 (Jackson ImmunoResearch #115-165-003, 1:500); Goat anti-mouse IgG FITC (Jackson ImmunoResearch # 115-095-003, 1:500).

### Adult fly infection and survival experiments
For bacterial infection, cell suspensions of freshly cultured *M. luteus* (OD600 = 1) and *S. aureus* (OD600 = 0.5) were used as working solutions. Adult male flies, aged 3–6 days, were anaesthetized with CO$_2$ and infected by being pricked in the thorax with a thin needle dipped previously into the bacterial working solution. Survival experiments were done at 29 °C. Surviving flies were transferred daily into fresh vials and counted.

### Cell transfection, western blot, immunoprecipitation, immunostaining, GST pull-down and quantitative real-time PCR (qRT-PCR)
Cells were transfected with Effectene transfection reagent (Qiagen). Western blot, immunoprecipitation and immunostaining were performed following standard protocols as described[88]. GST pull-down assay was performed as described[89]. qRT-PCR was carried out using CFX96 real-time system (Bio-Rad) as described[86]. The following qRT-PCR primers were used in this study: *CTGF*: 5'- CATCTTCGGTGGT ACGGTGT-3' and 5'-TTCCAGTCGGTAAGCCGC-3'; *AJUBA*: 5'-TACCAGG ACGAGCTAACAGC-3' and 5'-TACAGGTGCCGAAGTAGTCC-3'; *CYR61*: 5'- CGGGTTTCTTTCACAAGGCG-3' and 5'-TGAAGCGGCTCCCTGTTT TT-3'; *Cka*: 5'- ATACGGGTCCAGTTCTGTGC-3' and 5'-CCACAGCTT AACCGTTCCAT-3'; *Dpt*: 5'- ATCCTGATCCCCGAGAGATT-3' and 5'-C GTTGAGGCTCAGATCGAAT-3'; *Mtk*: 5'- TACATCAGTGCTGGCAGAG C-3' and 5'-AATAAATTGGACCCGGTCTTG-3'.

### In vitro kinase assay
In vitro kinase assay was performed as previously described[86]. Briefly, S2R+ cells were transfected with desired plasmids. 48 h post-transfection, cells were lysed and kinases expressed by the transfected plasmids were immunoprecipitated. The immunoprecipitated kinases were then incubated with bacterially purified recombinant GST-tagged substrates in kinase buffer at 37 °C for 30 min. The reaction mixtures were either directly subjected to SDS-PAGE analysis or further supplemented with 2.5 mM PNBM for 2 h at room temperature before analyzed via SDS-PAGE.

### Luciferase assay
Luciferase assay was performed as previously described[90]. Briefly, *Drosophila* S2R+ cells were seeded on 48-well plate. After 24 h, cells were transfected with the HRE luciferase reporter construct together with Renilla luciferase reporter plasmid (as the internal control) using Effectene transfection reagent (Qiagen). Luciferase assay was performed at 24 h post transfection by using Dual Luciferase Assay system (Promega) following the manufacturer's instructions and a FLUOstar Lumiometer (BMG Lab Technologies).

### Knockout cell line generation
CRISPR-Cas9 approach was used to generate *MAP3K7* knockout or *LATS1/2* double knockout HEK293T cells. gRNA sequences were designed through the web tool CHOPCHOP (https://chopchop.cbu.uib.no/) and the oligos were annealed before ligated into PX458 vector (Addgene 48138). The PX458-gRNA plasmids were then transfected into HEK293T cells with Effectene transfection reagent (Qiagen). 72 h later, transfected cells were trypsinized and suspended with PBS, and GFP-positive cells were then sorted into 96-well plates with one cell per well by flow cytometry. The sorted clones were further expanded and screened by western blot with primary antibodies against MAP3K7 or LATS1/2 before a final validation by Sanger sequencing.

### Hemocytes extraction
Extraction of larval hemocytes was carried out following the bleeding protocol as described[91]. Briefly, 3rd instar larvae were washed with sterile ddH$_2$O before transferred to droplets of PBS. Quickly ripped the outer cuticle with forceps and allowed the hemolymph to flow into the PBS. For adult hemocyte isolation, the anaesthetized adult flies were pierced in the thorax with a sterile needle and then put into a 500 μL Eppendorf tube with a pinhole on the bottom. The 500 μL tube was further placed into a 1.5 mL Eppendorf tube containing 75% Schneider's medium and 25% FBS. The hemolymph containing the hemocytes were collected by centrifuged at 5000 rpm for 5 min at 4 °C.

### Phagocytosis assay
200 μL of isolated larval hemolymph were incubated with ~2 × 10$^6$ dead *E. coli* expressing GFP first at 4 °C for 30 min and then at 26 °C for 1 h. Equal volume of 0.4% trypan blue was added before subjected for flow cytometry analysis. The phagocytic index was calculated following a formula as described[92].

### Statistics and reproducibility
All statistical analyses were done using GraphPad Prism 9.0.1 (GraphPad Software). No statistical method was used to predetermine sample size. Two-tailed unpaired Student's *t* test, one-way ANOVA and two-way ANOVA with Sidak's multiple comparisons test, Tukey's multiple comparisons test and Dunnett's multiple comparisons test were used for statistical analysis of data. Survival curves were plotted and analyzed by log-rank test. Detailed statistical analysis methods, sample size, or the number of biological replicates is indicated in the figure legends. The *p* values were labeled in the figures. Exact *p* values

ranging from 0.9999 to 0.0001 are reported as precise numbers, while those less than 0.0001 are indicated as <0.0001. The $p$ values lower than 0.05 were considered statistically significant.

## Reporting summary

Further information on research design is available in the Nature Portfolio Reporting Summary linked to this article.

## Data availability

All data supporting the findings of this study are available from the corresponding author upon request. The source data for Figs. 1A–K, M, 2, 3A–F, J–L, 4A–D, F–I, 5, 6, 7, and Supplementary Figs. 1, 2, 3A–C, G–J, 4, 5B, 6B, C, 7B, D, 8, 9 are provided as a Source Data file. This paper does not analyze any dataset or report original code. The *Drosophila* Interaction Database (DroID, http://www.droidb.org/) was used to identify protein interactions. Source data are provided with this paper.

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

## Acknowledgements

We would like to thank Dr. Dawang Zhou (Xiamen University) for GFP-*E. coli* bacteria, Dr. Qiao Wu (Xiamen University) for GFP-LC3 plasmid. We thank Bloomington Drosophila Stock Center (BDSC), Vienna Drosophila Resource Center (VDRC), Drosophila Genomics Resource Center (DGRC), and Developmental Studies Hybridoma Bank (DSHB) for fly strains and reagents. This study was supported in part by grants from National Natural Science Foundation of China (32170873 and 31970891 to B.L.), the National Key Research and Development Program of China (2019YFA0802002) and the NIH (EY015708 to D.P.). D.P. is an investigator of the Howard Hughes Medical Institute.

## Author contributions

Y.Z. and B.L. conceived the project. Y.Y., H.Z., X.H., C.W., K.Z., J.D., Y.Z., J.W., X.C., X.M., H.P., R.S., D.P. and B.L. designed experiments and analyzed data; Y.Y., H.Z., X.H., C.W., K.Z., J.D., Y.Z., J.W., X.C., X.M., H.P., R.S. and B.L. performed the experiments; B.L. supervised the study; B.L. wrote the manuscript with input from Y.Y., H.Z., X.H., Y.Z. and D.P.

## Competing interests

The authors declare no competing interests.
