## [Peer Review File · Nature Communications]

Innate immune and proinflammatory signals activate the Hippo pathway via a Tak1-STRIPAK-Tao axisREVIEWER COMMENTS

Reviewer #1 (Remarks to the Author):

The manuscript by Yang et. al, explores the molecular mechanisms underlying the activation of the Hippo signaling pathway by innate immune and proinflammatory signals. The researchers identified a signaling axis involving Tak1, a key kinase in innate immune and inflammatory signaling, that activates the Hippo pathway. They find that Tak1 induces the lysosomal degradation of Cka/STRN, an essential subunit of the STRIPAK PP2A complex, which is a suppressor of the Hippo signaling pathway. The suppression of the STRIPAK complex leads to the activation of Hippo signaling through the TAO-Hpo/MST signaling axis. The study shows that Tak1 can activate Hippo signaling by phosphorylating and activating key components of the pathway, including Hpo, Wts, Mats, and Yki. Finally, the researchers demonstrate that Tak1-induced Hippo signaling is involved in various cellular processes, such as cell death, phagocytosis, and innate immune memory. Overall, this research establishes a molecular connection between innate immune/inflammatory signaling and the Hippo signaling pathway. Given that their findings have several key implications for understanding the role of the Hippo pathway in infectious, inflammatory, and malignant diseases, this manuscript can be of interest for a wide audience and may offer insights for therapeutic interventions in these conditions. The study is well supported by biochemical, in vitro and in vivo data, yet there are a couple of data points that are weak and require further confirmation.

The increased expression of ban-lacZ upon Tak1 knockdown in figure 1L seems weak. Can this data be better shown in bigger pictures and maybe further supported by using another reporter such as Ex-lacZ or Kibra-lacZ?

Similarly, the differences in phagocytic index shown in figure 6A' and C' seem negligible. Is this biologically significant?

Lastly, I would like to see a bit more discussion of the biology of this pathway. How does this effect compare to the Toll regulation of Hippo signaling? Are the two signals of similar strength? How does the Tak1 pathway activity compare to the Toll activity in vivo? Do the

authors expect that the same target genes will be regulated by the two pathways? Is there synergy? Maybe it would be valuable to add some more data comparing Toll and Tak1 effects on target genes etc. The paper is very much focused on the biochemistry of the pathway but I think some more biology would increase the impact of this paper.

Reviewer #2 (Remarks to the Author):

Yang et al. report a detailed axis mediating Hippo pathway activation by innate immune and pro inflammatory signaling.

The study is well designed and the results have good quality.

Specific comments:

1) Please provide quantification in the following figures:

Figures 1A, B, C, D, E, F, G, H, I

Figures 2A, B, C, D, E, F, G, H, I, G

Figure 3A

Figures 4D, F

Figures 5A, B, C, D, E, F, G, M

Figure S3H

2) Data related to CKA degradation by proteasome and lysosome (Figure 3C and Figure S3C) are not convincing. I don't see any obvious difference supporting the conclusion that Tak1 induces degradation of CKA by the lysosomal pathway rather than the proteasomal pathway.

3) Figure 1L is not convincing

4) Figures 3G-I are not convincing. Staining is diffuse all over the cells.

5) Similarly Figure 4E not very convincing

Reviewer #3 (Remarks to the Author):

Liu and colleagues follow up on their 2016 Cell paper (involvement of the Hippo pathway in the humoral branch of innate immunity in fly fat tissue) to present a story about innate immune and proinflammatory signals promoting Hippo pathway activation via the Tak1-STRIPAK-Tao module in *Drosophila* macrophages, which might enhance the phagocytic activity of these cells. Proof-of-principle experiments are also shown in human cells. The signaling experiments are mostly well done, but their interpretation is sometimes not as clear as stated because of changes in protein levels that need to be better considered (please see point 1 below). Mechanistically, the authors suggest that Tak1 induces Cka degradation via the macroautophagy-lysosome pathway, but this aspect of the study is currently not properly supported, that is, standard tests of the autophagy field are missing (please see points 2-3 below). The potential role of this new signaling module in cell death is overinterpreted and further experiments are necessary to prove this (please see points 4-5 below). Currently we don't know why there are phagocytosis defects in flies either, because the number and identity of blood cells have not been studied despite Hippo signaling being a known regulator of blood cell differentiation/lineage specification (please see point 6). Lastly, the authors do not actually show that LPS or infection activates Hpo in fly blood cells (please see point 7). These caveats dramatically reduce my enthusiasm for the current version of the manuscript. I think that experimentally addressing these points would not only better support the model but also make this work interesting for a much broader audience, which could be a better match with the broad scope of Nature Communications.

Major comments:

1. Often the overall level of the studied proteins also decreases concomitant with the decrease of the phosphorylated form, see p-Yki and HA(Yki) in Fig1I, or both increase as in Fig7E-G. Along the same lines, the level of Flag-Cka strongly decreases upon myc-Tak1 expression, so it's no surprise that its amount is also lower in the IP of Fig3A (as the authors also notice and later analyze why) and Fig3D (Atg8a). Thus, regulation of protein level/stability can account for these changes, which do not support the claims of decreased phosphorylation/interaction. It is thus very important to quantify all western blots (from at least 3 biological replicates) and compare phosphorylation or IP changes to overall or input

protein levels, respectively.

2. It is not clear which lysosomal degradation pathway is involved in Cka degradation because BafA1 blocks the V-ATPase responsible for acidification (so all lysosomal degradation). Current data do not properly support that it is happening via the main autophagy pathway as pictured in Fig7N. To exclude the role of endosomal microautophagy or a variant of conjugation of Atg8 to single membranes (CASM), it is important to test knockdown/knockout of several autophagy genes (not only the commonly tested Atg5 or another one from Atg8 conjugation pathway genes, but also more macroautophagy-specific ones such as Atg14, Atg9, Atg2, Atg1 kinase complex subunits: FIP200, Atg13, Atg1/ULK1-2, Atg101), ideally in both Drosophila and human cells. I note here that the manuscript heavily relies on experiments with macrophage-like but highly heterogeneous and aneuploid S2 cells, so it would be preferable to also include animal data regarding this point.

3. It should be properly discussed that Tak1 is a known selective autophagy cargo degraded in lysosomes, and it has a LIR motif to bind Atg8a. Importantly, point mutation of the LIR in Tak1 prevents its selective degradation (PMID: 35081354 – ref. 43). But then how is Cka degraded if it is also targeted by autophagy? Does it also have a LIR motif? This is very easy to predict bioinformatically (note that LIR might be phosphorylation-dependent)? If yes, does this mediate its selective degradation (analyzed via LIR point mutations)? If not, is Cka degraded along with Tak1 (by forming a complex), so is Cka degradation blocked in these published Tak1 LIR mutants? These are very simple and routine experiments to carry out.

4. Line 396: “Hippo signaling is involved in Tak1-induced cell death”. The authors do not test cell death here, only document that the small irregular eye phenotype of Tak1 overexpression is partially suppressed by Wts RNAi (and to a small extent by Tao1 RNAi). For this claim (even appearing in the abstract), it is necessary to show (by anti-active caspase staining, TUNEL etc.) that suppression of Tak1-induced eye phenotype is indeed caused by decreased apoptosis (rather than for example compensatory hyperproliferation).

5. Does the block of autophagy affect the Tak1-induced rough eye phenotype and cell death? Please test RNAi lines against multiple Atg genes (see also point 2).

6. Proper analysis of hemocyte phenotypes is missing. At the minimum, the total number of blood cells should be established in flies in the various genotypes. For Fig5, the ratio of larval plasmatocyte (macrophage) vs lamellocyte numbers should be quantified because there will be insufficient phagocytosis if there are not enough phagocytes (macrophages), which is commonly caused by induction of lamellocyte differentiation. Along the same lines, adult blood cell numbers and morphology should be shown for Fig6, including *hpo/yki* rescue. Related papers that should be discussed are PMID: 25454587 (Hippo regulates hematopoiesis), PMID: 25454586 (Yorkie regulates Notch-dependent blood cell lineage specification).

7. Based on the authors model, Hpo is regulated in blood cells during infection. This should be analyzed using *bantam* or expanded as a readout in fly macrophages *in vivo* because the key message of the story: “LPS/infection induces *hpo* activation” is not actually shown now.

Minor comments:

8. The discussion is rather spartan. Please expand. E.g., how is phagocytosis regulated by this signaling? Also, it is not clear whether the crosstalk between Tak1/NFκB and Tak1/Hpo modules is only limited to the level of Tak1 or also observed more downstream. Does the knockdown of Kenny (another known selective autophagy cargo) produce the same phenotypes as Tak1, and can it be rescued by Hpo/Yki? This should be at least discussed.

9. Knockdown efficiencies need to be shown, preferably on the protein level if antibodies are available.

10. Fig2H: it cannot be excluded that other molecules bridge Cka and Tao1 because cell lysate is used in the pulldown. Please either repeat the pulldown experiment with recombinant Tao1 (or confirm by Y2H for example) or tone down/refrain from claiming direct interaction based on this experiment.

11. Atg8a is not really the *Drosophila* ortholog of LC3 proteins because it is evolutionarily closer to GABARAPs. It is better to call it an ortholog of yeast Atg8/mammalian Atg8 family

(both GABARAP and LC3) proteins. Atg8 proteins exist in two forms in cells: unlipidated (cytosolic) and lipidated (autophagic membrane associated). Please show both bands in Atg8a western blots (Fig3D,F).

12. Fig3G-I. Atg8a distribution should be punctate if it reflects autophagosome association (if not then this does not support autophagic degradation of Cka). Please show higher magnification/better resolution images. Is this colocalization seen in native macrophages of the fly as well?

13. Lines 343-345: The authors interpretation of “However, LPS failed to reduce the protein level of Cka in primary hemocytes isolated from Tak1 knockout flies (Figure 4F), suggesting the reduction of Cka level by LPS in hemocytes is Tak1-dependent.” is ignoring the fact that the level of endogenous Cka is always much lower in Tak1 mutant cells than in WT (not only Ctrl but also LPS condition). Please discuss this properly.

14. Please show the level of all proteins in the inputs in immunoprecipitation experiments, as some are only shown in IPs now.

15. For consistency, please also show how MAP3K7 OE affects CYR61 mRNA expression, and how MAP3K7 KO affects AJUBA mRNA levels in FigS7.

16. Please specify which LC3 protein is studied in Fig7K,M (is it LC3B?)

Reviewer #1:

The manuscript by Yang et. al, explores the molecular mechanisms underlying the activation of the Hippo signaling pathway by innate immune and proinflammatory signals. The researchers identified a signaling axis involving Tak1, a key kinase in innate immune and inflammatory signaling, that activates the Hippo pathway. They find that Tak1 induces the lysosomal degradation of Cka/STRN, an essential subunit of the STRIPAK PP2A complex, which is a suppressor of the Hippo signaling pathway. The suppression of the STRIPAK complex leads to the activation of Hippo signaling through the TAO-Hpo/MST signaling axis. The study shows that Tak1 can activate Hippo signaling by phosphorylating and activating key components of the pathway, including Hpo, Wts, Mats, and Yki. Finally, the researchers demonstrate that Tak1-induced Hippo signaling is involved in various cellular processes, such as cell death, phagocytosis, and innate immune memory. Overall, this research establishes a molecular connection between innate immune/inflammatory signaling and the Hippo signaling pathway. Given that their findings have several key implications for understanding the role of the Hippo pathway in infectious, inflammatory, and malignant diseases, this manuscript can be of interest for a wide audience and may offer insights for therapeutic interventions in these conditions. The study is well supported by biochemical, in vitro and in vivo data, yet there are a couple of data points that are weak and require further confirmation

We thank the reviewer for recognizing the novelty and quality of our study. Below we address the reviewer's specific points.

1. The increased expression of ban-lacZ upon Tak1 knockdown in figure 1L seems weak. Can this data be better shown in bigger pictures and maybe further supported by using another reporter such as Ex-lacZ or Kibra-lacZ?

We thank the reviewer for the helpful suggestion. To address the reviewer's concern, we further optimized the staining protocol mainly by adjusting the blocking time and antibody concentration. Now we obtained better staining results of *ban-lacZ* upon Tak1 knockdown than those in the original Figure 1L, and therefore replaced the original Figure 1L with a new picture in the revision. We further quantified the *ban-lacZ* intensity of the posterior compartment over that of the anterior compartment of the wing disc using ImageJ software. The results showed significant induction of *ban-lacZ* upon Tak1 knockdown. The quantification data is added as Figure 1L' in the revision. (also shown below as **Author response Figure 1** for the reviewer's convenience). Meanwhile, we also tested another reporter line for Hippo pathway, *ex-lacZ*, as suggested by the reviewer. The results showed that Tak1 knockdown promoted *ex-lacZ* expression as well (**Author response Figure 2**).

Author response Figure 1. Third instar wing discs expressing UAS-GFP only (top panel) or UAS-GFP plus UAS-Tak1 RNAi (bottom panel) in the posterior compartment by the *en*-Gal4 driver stained for lacZ expression (red). Note the increased expression of *ban-lacZ* upon Tak1 knockdown. Scale bars, 50 μ m. Quantification of *ban-lacZ* signal is shown in (L'). Data were analyzed using two-tailed Student's *t*-test and presented as mean \pm SD, *n*=9.

Author response Figure 2. Third instar wing discs expressing UAS-GFP only (top panel) or UAS-GFP plus UAS-Tak1 RNAi (bottom panel) in the posterior compartment by the *en-Gal4* driver stained for lacZ expression (red). Note the increased expression of *ex-lacZ* upon Tak1 knockdown.

2. Similarly, the differences in phagocytic index shown in figure 5A' and C' seem negligible. Is this biologically significant?

Thanks for the comments. The mild effect may be partly due to the incomplete nature of RNAi. Nonetheless, statistical analysis of at least three biological replicates shows significant difference in phagocytic index between control hemocytes and Tak1 knockdown hemocytes (as shown in Figure 5A', $p=0.0037$), and significant difference between Tak1 knockdown hemocytes and Tak1/Yki double knockdown hemocytes (as shown in Figure 5C', $p=0.0021$).

3. Lastly, I would like to see a bit more discussion of the biology of this pathway. How does this effect compare to the Toll regulation of Hippo signaling? Are the two signals of similar strength? How does the Tak1 pathway activity compare to the Toll activity in vivo? Do the authors expect that the same target genes will be regulated by the two pathways? Is there synergy? Maybe it would be valuable to add some more data comparing Toll and Tak1 effects on target genes etc. The paper is very much focused on the biochemistry of the pathway but I think some more biology would increase the impact of this paper.

We thank the reviewer for bringing up these appealing questions. Answering these questions would definitely improve the impact of our study. *Drosophila* innate immune system is composed of the humoral response which is mainly mediated by fat body through secretion of antimicrobial peptides, and the cellular response which is mainly mediated by hemocytes through phagocytosis, melanization and encapsulation of the pathogens^{1,2}. Phagocytosis, melanization and encapsulation of the pathogens are mediated by plasmatocytes, crystal cells, and lamellocytes, respectively. Our previous study³ and this study collectively showed that both Toll-Hippo axis and Tak1-Hippo axis operate in fly S2 cells, a phagocytic haematopoietic cell line which has similar phagocytic ability to the primary phagocytes^{4,5}, suggesting that both axis may function in hemocytes. In contrast, only Toll-Hippo axis operates in the fat body³. Therefore, the hemocytes rather than fat body is a good model to compare the effect of the two signaling axes in a physiological context. By comparing the Hippo signaling activity in Toll mutant hemocytes with that in Tak1 mutant hemocytes will tell the relative contribution of the two pathways to the Hippo signaling. Meanwhile, by comparing the effect of Toll/Tak1 double mutant hemocytes with Toll or Tak1 single mutant hemocytes would answer whether these two pathways have synergistic effect in regulating the Hippo signaling.

Our current study identified a critical role for Tak1-Hippo axis in the phagocytosis of the hemocytes. However, the role of Toll-Hippo axis in hemocyte physiology remains unknown. To answer the question whether we expect that the same target genes will be regulated by Toll-Hippo axis and Tak1-Hippo axis, we should determine the role of Toll-Hippo axis in hemocyte physiology first. If the Toll-Hippo axis also controls phagocytosis like the Tak1-Hippo axis does, we would expect that these two signaling axis likely controls a large number of the same target genes, and vice versa. Moreover, this question can be answered by comparing the transcriptome of Toll or Tak1 mutant hemocytes through RNA sequencing. We have discussed these important questions raised by the reviewer in the revision (**Lines 541-550**).

Reviewer #2:

Yang et al. report a detailed axis mediating Hippo pathway activation by innate immune and pro inflammatory signaling.

The study is well designed and the results have good quality.

We thank the reviewer for recognizing the quality of our data. Below we address the reviewer's specific points.

Specific comments:

1. Please provide quantification in the following figures:

Figures 1A, B, C, D, E, F, G, H, I

Figures 2A, B, C, D, E, F, G, H, I, G

Figure 3A

Figures 4D, F

Figures 7A, B, C, D, E, F, G, M

Figure S3H

We have quantified these figures with ImageJ software and included the quantification results in the revised figures.

2. Data related to CKA degradation by proteasome and lysosome (Figure 3C and Figure S3C) are not convincing. I don't see any obvious difference supporting the conclusion that Tak1 induces degradation of CKA by the lysosomal pathway rather than the proteasomal pathway.

We have performed statistical analysis of three biological replicates to compare the degradation of Cka between DMSO (vehicle control) and BafA1- (inhibitor of lysosomal pathway) treated groups, or DMSO and PS-341- (inhibitor of proteasomal pathway) treated groups. As shown in **Author response Figure 3**, Tak1-induced Cka degradation was significantly inhibited by BafA1 ($p=0.0044$), but not PS-341 ($p=0.6559$), which supports the conclusion that Tak1 mediates Cka degradation through the lysosomal pathway rather than the proteasomal pathway.

Author response Figure 3. The protein level of Cka was quantified using ImageJ software and the degradation curve was generated based on the quantification result. Note that the degradation of Cka was significantly suppressed by BafA1, but not PS-341. The degradation curves were analyzed using Two-Way ANOVA.

3. *Figure 1L is not convincing.*

We thank the reviewer for pointing out the limitation of this figure as well as figures in the following points 4 and 5. For Figure 1L, we have further optimized the staining protocol mainly by adjusting the blocking time and antibody concentration. Now we obtained better staining results of *ban-lacZ* upon Tak1 knockdown than the original Figure 1L. We have replaced the original Figure 1L with a new picture in the revision. Moreover, we quantified the signal intensity of *ban-lacZ* using ImageJ software and compared the results of the posterior compartment with those of the anterior compartment of the wing disc. Statistical analysis shows significant induction of *ban-lacZ* in the posterior compartment upon Tak1 knockdown. The quantification data is added as Figure 1L' in the revision. (also shown in **Author response Figure 1** for the reviewer's convenience). Meanwhile, we also tested another reporter line for Hippo pathway, *ex-lacZ*. The results showed that Tak1 knockdown promoted *ex-lacZ* level as well (**Author response Figure 2**).

4. *Figures 3G-I are not convincing. Staining is diffuse all over the cells.*

Indeed, Atg8 proteins form puncta due to translocation to newly formed autophagosomes upon autophagy induction⁶. However, we were not able to observe the punctate pattern of the Atg8 in **fixed** *Drosophila* S2R+ cell line even under typical conditions for autophagy induction such as nutrient starvation, suggesting that Atg8 puncta may be compromised during the fixation process of S2R+ cells. Instead, we generated fluorescent protein-tagged constructs for the related proteins (Atg8a, Cka, Tak1, Tak1^{S176A}), and repeated this experiment in **live** S2R+ cells in the revision. By doing so, we clearly observed Atg8a puncta. More importantly, Tak1, but not Tak1^{S176A}, promoted the colocalization of Cka and Atg8a puncta. These improved results provide better evidence supporting our original conclusion. We have therefore updated Figure 3G-I by replacing the original pictures of fixed cells with improved pictures of live cells in the revision. (also shown below as **Author response Figure 4** for the reviewer's convenience)

Author response Figure 4. Immunostaining performed in live S2R+ cells showing enhanced colocalization of Cka and Atg8a upon co-expression of Tak1, but not Tak1^{S176A}. Arrow heads point to representative puncta with protein colocalization. Scale bars, 4 μ m.

5. Similarly Figure 4E not very convincing.

In the revision, we have further optimized the staining protocol mainly by adjusting the blocking time and antibody concentration. Now we obtained better staining results than the original Figure 4E. We have replaced the original Figure 4E with a new picture in the revision. (also shown below as **Author response Figure 5** for the reviewer's convenience)

4E

Author response Figure 5. Immunostaining was performed in S2R+ cells transfected with Flag-Cka and Lamp1-GFP constructs, showing enhanced colocalization of Cka and Lamp1 upon LPS treatment (10 μ g/mL, 3 h). Arrow heads are representative Cka puncta colocalized with Lamp1-GFP. Scale bars, 5 μ m.

Reviewer #3:

Liu and colleagues follow up on their 2016 Cell paper (involvement of the Hippo pathway in the humoral branch of innate immunity in fly fat tissue) to present a story about innate immune and proinflammatory signals promoting Hippo pathway activation via the Tak1-STRIPAK-Tao module in Drosophila macrophages, which might enhance the phagocytic activity of these cells. Proof-of-principle experiments are also shown in human cells. The signaling experiments are mostly well done, but their interpretation is sometimes not as clear as stated because of changes in protein levels that need to be better considered (please see point 1 below). Mechanistically, the authors suggest that Tak1 induces Cka degradation via the macroautophagy-lysosome pathway, but this aspect of the study is currently not properly supported, that is, standard tests of the autophagy field are missing (please see points 2-3 below). The potential role of this new signaling module in cell death is overinterpreted and further experiments are necessary to prove this (please see points 4-5 below). Currently we don't know why there are phagocytosis defects in flies either, because the number and identity of blood cells have not been studied despite Hippo signaling being a known regulator of blood cell differentiation/lineage specification (please see point 6). Lastly, the authors do not actually show that LPS or infection activates Hpo in fly blood cells (please see point 7). These caveats dramatically reduce my enthusiasm for the current version of the manuscript. I think that experimentally addressing these points would not only better support the model but also make this work interesting for a much broader audience, which could be a better match with the broad scope of Nature Communications.

We thank the reviewer for recognizing the quality of most of our data, and for pointing out the limitation of the current version of the manuscript. Below we address the reviewer's specific points.

Major comments:

1. Often the overall level of the studied proteins also decreases concomitant with the decrease of the phosphorylated form, see p-Yki and HA(Yki) in FigII, or both increase as in Fig7E-G. Along the same lines, the level of Flag-Cka strongly decreases upon myc-Tak1 expression, so it's no surprise that its amount is also lower in the IP of Fig3A (as the authors also notice and later analyze why) and Fig3D (Atg8a). Thus, regulation of protein level/stability can account for these changes, which do not support the claims of decreased phosphorylation/interaction. It is thus very important to quantify all western blots (from at least 3 biological replicates) and compare phosphorylation or IP changes to overall or input protein levels, respectively.

Indeed, sometimes the level of a transiently expressed protein varies when co-transfected with another plasmid and this phenomenon is likely due to resource competition among co-transfected plasmids⁷ or promoter interference⁸. In the revision, we first tried to

minimize the protein level variations through strategies such as 1) using alternative promoters to drive the expression of the cDNA, 2) adjusting the proportion of the co-transfected plasmids, 3) adjusting the ratio between the amount of transfected plasmid and the amount of transfection reagents, and updated Figure 1I, 7E-G and 3D. Then to better support our conclusions, we have quantified all western blots and showed the mean of three independent experiments in the revised figures as suggested by the reviewer.

2. It is not clear which lysosomal degradation pathway is involved in Cka degradation because BafA1 blocks the V-ATPase responsible for acidification (so all lysosomal degradation). Current data do not properly support that it is happening via the main autophagy pathway as pictured in Fig7N. To exclude the role of endosomal microautophagy or a variant of conjugation of Atg8 to single membranes (CASM), it is important to test knockdown/knockout of several autophagy genes (not only the commonly tested Atg5 or another one from Atg8 conjugation pathway genes, but also more macroautophagy-specific ones such as Atg14, Atg9, Atg2, Atg1 kinase complex subunits: FIP200, Atg13, Atg1/ULK1-2, Atg101), ideally in both Drosophila and human cells. I note here that the manuscript heavily relies on experiments with macrophage-like but highly heterogeneous and aneuploid S2 cells, so it would be preferable to also include animal data regarding this point.

Thanks for the constructive suggestion. To further validate that the macroautophagy pathway is required for Tak1/MAP3K-mediated Cka/STRN degradation, we first tested several macroautophagy genes including Atg1, Atg2, or Atg9 in S2 cells. Similar to Atg8, Tak1-induced Cka degradation was inhibited by Atg1, Atg2, or Atg9 knockdown in S2 cells. In line with the results in S2 cells, we also observed reduced Cka level upon Tak1 overexpression driven by hemocyte-specific Hml-Gal4 driver, which was partially rescued by Atg1, Atg9, Atg13 or Atg101 knockdown in *Drosophila*. Consistent with the findings in *Drosophila*, MAP3K7-induced STRN degradation was also suppressed by ULK-101, a potent inhibitor of ULK1/2 (mammalian Atg1), in human HEK293T cells. Altogether, these data strongly suggest that the macroautophagy pathway is required for Tak1/MAP3K7-mediated Cka/STRN degradation in both *Drosophila* and mammalian cells. These data are added as **Figure S3I, Figure 4F and Figure S9B** in the revision. (also shown below as **Author response Figure 6, 7 and 8** for the reviewer's convenience).

Author response Figure 6. S2 cells pretreated with dsRNAs of Luciferase or several Atg proteins and then transfected with indicated plasmids were treated with CHX (50 $\mu\text{g}/\text{mL}$) for indicated times. Note that the accelerated degradation of Cka upon Tak1 co-expression was impeded by Atg1, Atg2 or Atg9 knockdown.

Author response Figure 7. Cka level was examined in stage 17 fly embryos with indicated genotype via western blotting. Note that the reduced level of Cka resulted from Tak1 overexpression was partially rescued by Atg1, Atg9, Atg13 or Atg101 knockdown.

Author response Figure 8. HEK293T cells transfected with indicated plasmids were pretreated with ULK-101 (5 μM) for 1 h before CHX treatment for the indicated time. Note that MAP3K7-induced STRN1 degradation was suppressed by ULK-101 treatment.

3. It should be properly discussed that *Tak1* is a known selective autophagy cargo degraded in lysosomes, and it has a LIR motif to bind *Atg8a*. Importantly, point mutation of the LIR in *Tak1* prevents its selective degradation (PMID: 35081354 – ref. 43). But then how is *Cka* degraded if it is also targeted by autophagy? Does it also have a LIR motif? This is very easy to predict bioinformatically (note that LIR might be phosphorylation-dependent)? If yes, does this mediate its selective degradation (analyzed via LIR point mutations)? If not, is *Cka* degraded along with *Tak1* (by forming a

complex), so is Cka degradation blocked in these published Tak1 LIR mutants? These are very simple and routine experiments to carry out.

Thank you for the constructive suggestion. Three conserved LIRs have been identified in Cka using the LIR prediction software iLIR⁹, and the LIR2 (aa. 313-318; ANFEFL) is crucial for Atg8a binding¹⁰. To answer the reviewer's question whether the LIR mediates Cka degradation, we disrupted the LIR2 of Cka either by substituting the phenylalanine (F) and leucine (L) with alanine (A) as described¹⁰, or by deleting the entire LIR2 motif. Results indicated that Cka mutants with defective LIR2 had comparable degradation kinetics as wild-type Cka, suggesting that LIR2-mediated Atg8a binding is not essential for Cka degradation.

On the other hand, two LIRs have also been identified in Tak1 which mediate Tak1-Atg8a interaction and the selective degradation of Tak1¹¹. Since our model argues that Tak1 acts as a cargo receptor for the selective degradation of Cka, we would expect Tak1-mediated Cka degradation to be dependent on Tak1's LIRs. To test this prediction, we generated Tak1 mutant with inactivated LIR1, the predominant LIR that mediates Tak1-Atg8 association, as previously described¹¹, and examined its effect on Cka degradation. The result indicated that inactivating LIR1 indeed impaired Tak1's ability of mediating Cka degradation, which further corroborated our model. These data are added as **Figure S3J** and **Figure 3L**, respectively, in the revision. (also shown below as **Author response Figure 9 & 10** for the reviewer's convenience).

S3J

Author response Figure 9. S2R+ cells were transfected with indicated plasmids and treated with CHX (50 μg/mL) for designated times. Note the comparable degradation kinetics of Cka, Cka^{mutLIR2} and Cka^{ΔLIR2}.

3L

Author response Figure 10. S2R+ cells were transfected with indicated plasmids and treated with CHX (50 µg/mL) for designated times. Note that the degradation of Cka by co-transfection of Tak1^{mutLIR1} was slower than that of Tak1.

4. Line 396: “Hippo signaling is involved in Tak1-induced cell death”. The authors do not test cell death here, only document that the small irregular eye phenotype of Tak1 overexpression is partially suppressed by Wts RNAi (and to a small extent by Tao1 RNAi). For this claim (even appearing in the abstract), it is necessary to show (by anti-active caspase staining, TUNEL etc.) that suppression of Tak1-induced eye phenotype is indeed caused by decreased apoptosis (rather than for example compensatory hyperproliferation)

We stained the larval eye imaginal discs for cleaved Dcp-1, an effector caspase in *Drosophila*, to monitor apoptotic cell death. Consistent with the adult eye size phenotype, we observed that Tak1 overexpression resulted in increased number of cleaved Dcp-1 puncta, which was largely inhibited by concurrent Tao-1 or Wts knockdown. These data support our conclusion that Hippo signaling is involved in Tak1-induced cell death. These data are added as **Figure S7C** and **S7D** in the revision. (also shown below as **Author response Figure 11** for the reviewer’s convenience, the Atg RNAi pictures are for addressing the following point 5).

S7C

S7D

Author response Figure 11. (S7C). Eye discs dissected from 3rd instar larvae with indicated genotypes were stained for cleaved Dcp-1. Note that the increased number of cleaved Dcp-1 puncta from Tak1 overexpression was partially suppressed by simultaneous depletion of Tao-1, Wts, Atg1, Atg9, Atg13 or Atg101. Scale bar, 50 μ m. (S7D). Quantification of cleaved Dcp-1 puncta number in (S7C). Data shown are mean \pm SD, (*sev*-Gal4: *n*=16; *sev*>Tak1: *n*=28; *sev*>Tak1;*wts*.RNAi: *n*=7; *sev*>Tak1;*Tao-1*.RNAi: *n*=7; *sev*>Tak1;*Atg1*.RNAi: *n*=10; *sev*>Tak1;*Atg9*.RNAi: *n*=15; *sev*>Tak1;*Atg13*.RNAi: *n*=8; *sev*>Tak1;*Atg101*.RNAi: *n*=6), two-tailed Student's *t*-test.

5. Does the block of autophagy affect the Tak1-induced rough eye phenotype and cell death? Please test RNAi lines against multiple Atg genes (see also point 2).

As suggested by the reviewer, we tested multiple Atg genes to examine whether blocking autophagy would affect the Tak1-induced rough eye phenotype and cell death. The results showed that depletions of multiple Atg genes such as Atg1, Atg9, Atg13 and Atg101 all suppressed Tak1-induced small eye phenotype (**Figure S7A & S7B**) and apoptotic cell death (**Figure S7C & S7D**), which is consistent with our model that autophagy pathway is required for Tak1-mediated Cka degradation and Hippo pathway activation. Adult eye data are updated in the original Figure S6. Active Dcp-1 staining results are added as **Figure S7C and S7D** in the revision. (Adult eye data are also shown below as **Author response Figure 12** for the reviewer's convenience; Dcp-1 staining results are shown above as **Author response Figure 11**).

S7A

S7B

Author response Figure 12. (S7A). Adult eye images of the indicated genotypes, all taken under the same magnification. Note that the reduced eye size resulted from Tak1 overexpression was partially rescued by simultaneous depletion of Tao-1, Wts, Atg1, Atg9, Atg13 or Atg101. Scale bar, 100 μ m. (S7B). Quantification of eye size in (S7A). Data shown are mean \pm SD, (*sev*-Gal4: *n*=11; *sev*>Tak1: *n*=16; *sev*>Tak1;*wts*.RNAi: *n*=14; *sev*>Tak1;*Tao-1*.RNAi: *n*=11; *sev*>Tak1;*Atg1*.RNAi: *n*=13; *sev*>Tak1;*Atg9*.RNAi: *n*=11; *sev*>Tak1;*Atg13*.RNAi: *n*=12; *sev*>Tak1;*Atg101*.RNAi: *n*=9), two-tailed Student's *t*-test.

6. Proper analysis of hemocyte phenotypes is missing. At the minimum, the total number of blood cells should be established in flies in the various genotypes. For Fig5, the ratio of larval plasmatocyte (macrophage) vs lamellocyte numbers should be quantified because there will be insufficient phagocytosis if there are not enough phagocytes (macrophages), which is commonly caused by induction of lamellocyte differentiation. Along the same lines, adult blood cell numbers and morphology should be shown for Fig6, including hpo/yki rescue. Related papers that should be discussed are PMID: 25454587 (Hippo regulates hematopoiesis), PMID: 25454586 (Yorkie regulates Notch-dependent blood cell lineage specification).

During *Drosophila* haematopoiesis, the blood progenitor cells can give rise to three types of differentiated hemocytes, namely plasmatocyte, crystal cell and lamellocyte¹². The plasmatocyte is the professional phagocyte and is the most abundant cell type comprising ~95% of the hemocyte repertoire. The crystal cells represent ~5% of hemocytes in circulation, while the lamellocytes are barely observed in healthy animal but can be induced to differentiate by parasitic wasp infestation¹³. For Figure 5, we agree with the reviewer that changes in the identity of the blood cells would potentially contribute to the results. However, given that the Hml-Gal4 driver that we used in these figures are not expressed in blood progenitor cells¹⁴, it is unlikely that manipulating gene expression with this driver would affect the hemocyte differentiation. To test this possibility, we performed Giemsa staining of larval hemolymph smears following a described protocol¹⁵, and counted the numbers of differential hemocytes. The result indicated that the proportion of the plasmatocytes was comparable among the flies related to Figure 5. This result ruled out the possibility that the observed differential phagocytic ability was due to the change of the hemocyte identity. These data are added as **Figure S5** in the revision. (also shown below as **Author response Figure 13** for the reviewer's convenience). The related paper PMID: 25454587 and PMID: 25454586 are also cited in the revised manuscript (**Lines 406-412**).

For Figure 6, we determined the total numbers of the hemocytes in the related flies using a hemocytometer. The result indicated that the densities of the hemocytes were comparable among these flies, which ruled out the possibility that the varied innate immune memory phenotype was due to the change of the hemocyte number. These data are added as **Figure 6F** in the revision. (also shown below as **Author response Figure 14** for the reviewer's convenience). Moreover, we also examined the morphology of adult hemocytes as suggested by the reviewer. Plasmatocytes and crystal cells, but no lamellocytes, were observed in adult flies, which is consistent with the reported results¹⁶. Overall, we didn't observe obvious morphology difference of the hemocytes in the flies related to Figure 6. These data are also shown in **Author response Figure 15**.

S5

Author response Figure 13. The percentage of plasmatocytes, crystal cells and lamellocytes were shown as 100% stacked bar chart. Note the comparable proportion of plasmatocytes in the larvae of indicated genotype. The percentage of the plasmatocyte was analyzed using Two-way ANOVA with Dunnett's multiple comparisons test and shown as mean+s.d., $n=3$.

6F

Author response Figure 14. The relative hemocyte number of the flies with indicated genotype was determined via hemocytometer. Data were analyzed using one-way ANOVA with Tukey's honest significant difference (HSD) test and presented as mean± s.d., $n=5$.

Author response Figure 15. Adult hemocyte morphology of flies with indicated genotype. Scale bars, 5 μ m.

7. Based on the authors model, *Hpo* is regulated in blood cells during infection. This should be analyzed using *bantam* or *expanded* as a readout in fly macrophages *in vivo* because the key message of the story: “LPS/infection induces *hpo* activation” is not actually shown now.

As suggested by the reviewer, we monitored the expression of *expanded* (*ex*) in fly primary hemocytes upon LPS treatment via qRT-PCR. Result showed that LPS treatment led to reduced expression of *ex*, suggesting Hippo signaling activation upon LPS treatment in primary hemocytes. These data are added as **Figure 4H** in the revision. (also shown below as **Author response Figure 16** for the reviewer’s convenience).

Author response Figure 16. qRT-PCR assay performed in primary hemocytes showing that mRNA level of *ex* was reduced upon LPS treatment (10 μ g/mL, 0.5 h). Data were analyzed using two-tailed Student's t-test and presented as mean \pm SD, $n = 3$.

Minor comments:

8. *The discussion is rather spartan. Please expand. E.g., how is phagocytosis regulated by this signaling? Also, it is not clear whether the crosstalk between Tak1/NFkB and Tak1/Hpo modules is only limited to the level of Tak1 or also observed more downstream. Does the knockdown of Kenny (another known selective autophagy cargo) produce the same phenotypes as Tak1, and can it be rescued by Hpo/Yki? This should be at least discussed.*

Thanks for the suggestion. We have expanded the discussion regarding how phagocytosis is regulated by Tak1-mediated Hippo signaling (**Lines 515-526**). To evaluate whether Kenny has a similar effect as Tak1 on Hippo signaling, we examined whether Kenny would affect the phosphorylation of key components of the Hippo signaling. As can be seen in “**Author Response Figure 17**”, Kenny failed to induce the phosphorylation of major components of the Hippo pathway. This result indicates that Kenny is not involved in Hippo signaling regulation. Therefore, knockdown of Kenny is highly unlikely to produce the same phenotypes as Tak1. This point is also discussed in the revision (**Lines 555-557**).

Author response Figure 17. S2R⁺ cells were transfected with indicated plasmids. Phosphorylation of Tao1, Hpo, Mats and Yki was monitored through western blotting. Note that co-transfection of Kenny failed to induce the phosphorylation of Tao1, Hpo, Mats or Yki.

9. *Knockdown efficiencies need to be shown, preferably on the protein level if antibodies are available.*

We assessed the RNAi knockdown efficiency for all related figures either by quantified western blot or qRT-PCR (if antibody is not available). This information is added as **Table S1** in the revision.

10. *Fig2H: it cannot be excluded that other molecules bridge Cka and Tao1 because cell lysate is used in the pulldown. Please either repeat the pulldown experiment with recombinant Tao1 (or confirm by Y2H for example) or tone down/refrain from claiming direct interaction based on this experiment.*

We toned down the statement in the revision (**Lines 236-240**).

4. *Atg8a is not really the Drosophila ortholog of LC3 proteins because it is evolutionarily closer to GABARAPs. It is better to call it an ortholog of yeast Atg8/mammalian Atg8 family (both GABARAP and LC3) proteins. Atg8 proteins exist in two forms in cells: unlipidated (cytosolic) and lipidated (autophagic membrane associated). Please show both bands in Atg8a western blots (Fig3D,F).*

As suggested by the reviewer, we added GABARAP in the text (**Line 285**). We also showed both unlipidated and lipidated bands of Atg8 in Figure 3D & F. However, likely because the cells used in these experiments were not induced to undergo autophagy with

typical conditions such as nutrient starvation, the level of lipidated Atg8a band is very weak.

11. Fig3G-I. Atg8a distribution should be punctate if it reflects autophagosome association (if not then this does not support autophagic degradation of Cka). Please show higher magnification/better resolution images. Is this colocalization seen in native macrophages of the fly as well?

Indeed, Atg8 proteins form puncta due to translocation to newly formed autophagosomes upon autophagy induction ⁶. However, we were not able to observe the punctate pattern of the Atg8 in **fixed** *Drosophila* S2R+ cell line even under typical condition for autophagy induction such as nutrient starvation, suggesting that Atg8 puncta may be compromised during the fixation process of S2R+ cells. Instead, we generated fluorescent protein-tagged constructs for the related proteins (Atg8a, Cka, Tak1, Tak1^{S176A}), and repeated this experiment in **live** S2R+ cells in the revision. By doing so, we clearly observed Atg8a puncta. More importantly, Tak1, but not Tak1^{S176A}, promoted the colocalization of Cka and Atg8a puncta. These improved results provide better evidence supporting our original conclusion. We have therefore updated Figure 3G-I by replacing the original result pictures of fixed cells with improved pictures of live cells in the revision. (also shown as **Author response Figure 4** for the reviewer's convenience).

At this moment, we lack the required transgenic fly lines expressing fluorescent protein-tagged Atg8a, Cka, Tak1 or Tak1^{S176A} for live imaging to answer the question “*Is this colocalization seen in native macrophages of the fly as well?*” However, we hope the reviewer would agree with us that this specific question won't affect the main conclusion of our study. Given the time restriction of revision, we also hope the reviewer would agree with us that this question is better suited for a future study.

12. Lines 343-345: The authors interpretation of “However, LPS failed to reduce the protein level of Cka in primary hemocytes isolated from Tak1 knockout flies (Figure 4F), suggesting the reduction of Cka level by LPS in hemocytes is Tak1-dependent.” is ignoring the fact that the level of endogenous Cka is always much lower in Tak1 mutant cells than in WT (not only Ctrl but also LPS condition). Please discuss this properly.

The reviewer made the correct observation that the level of endogenous Cka is lower in Tak1 mutant cells than that in WT. This is likely because the homeostatic protein levels of endogenous Cka is also controlled by other mechanisms besides Tak1, such as Pelle, as shown in our previous paper ³. We have discussed this point in the revised manuscript (**Lines 376-379**).

13. Please show the level of all proteins in the inputs in immunoprecipitation experiments, as some are only shown in IPs now.

All proteins in the inputs are shown in the revision.

14. For consistency, please also show how MAP3K7 OE affects CYR61 mRNA expression, and how MAP3K7 KO affects AJUBA mRNA levels in FigS7.

Both CYR61 and AJUBA expressions are added in the revised Figure S8.

15. Please specify which LC3 protein is studied in Fig7K,M (is it LC3B?)

Yes, it is LC3B. We have added this information in Figure 7K & M.

References

1. Hultmark D. Drosophila immunity: paths and patterns. *Current opinion in immunology* **15**, 12-19 (2003).
2. Lemaitre B, Hoffmann J. The host defense of *Drosophila melanogaster*. *Annual review of immunology* **25**, 697-743 (2007).
3. Liu B, Zheng Y, Yin F, Yu J, Silverman N, Pan D. Toll Receptor-Mediated Hippo Signaling Controls Innate Immunity in *Drosophila*. *Cell* **164**, 406-419 (2016).
4. Ramet M, *et al.* *Drosophila* scavenger receptor Ci is a pattern recognition receptor for bacteria. *Immunity* **15**, 1027-1038 (2001).
5. Ramet M, Manfruelli P, Pearson A, Mathey-Prevot B, Ezekowitz RA. Functional genomic analysis of phagocytosis and identification of a *Drosophila* receptor for *E. coli*. *Nature* **416**, 644-648 (2002).
6. Kabeya Y, *et al.* LC3, a mammalian homologue of yeast Apg8p, is localized in autophagosome membranes after processing. *The EMBO journal* **19**, 5720-5728 (2000).
7. Di Blasi R, Marbiah MM, Siciliano V, Polizzi K, Ceroni F. A call for caution in analysing mammalian co-transfection experiments and implications of resource competition in data misinterpretation. *Nature communications* **12**, 2545 (2021).

8. Huliak I, Sike A, Zencir S, Boros IM. The objectivity of reporters: interference between physically unlinked promoters affects reporter gene expression in transient transfection experiments. *DNA and cell biology* **31**, 1580-1584 (2012).
9. Kalvari I, *et al.* iLIR: A web resource for prediction of Atg8-family interacting proteins. *Autophagy* **10**, 913-925 (2014).
10. Neisch AL, Neufeld TP, Hays TS. A STRIPAK complex mediates axonal transport of autophagosomes and dense core vesicles through PP2A regulation. *The Journal of cell biology* **216**, 441-461 (2017).
11. Tsapras P, *et al.* Selective autophagy controls innate immune response through a TAK1/TAB2/SH3PX1 axis. *Cell reports* **38**, 110286 (2022).
12. Meister M, Lagueux M. Drosophila blood cells. *Cellular microbiology* **5**, 573-580 (2003).
13. Lanot R, Zachary D, Holder F, Meister M. Postembryonic hematopoiesis in Drosophila. *Developmental biology* **230**, 243-257 (2001).
14. Jung SH, Evans CJ, Uemura C, Banerjee U. The Drosophila lymph gland as a developmental model of hematopoiesis. *Development* **132**, 2521-2533 (2005).
15. Rajak P, Dutta M, Roy S. Altered differential hemocyte count in 3rd instar larvae of *Drosophila melanogaster* as a response to chronic exposure of Acephate. *Interdisciplinary toxicology* **8**, 84-88 (2015).
16. Boulet M, Renaud Y, Lapraz F, Benmimoun B, Vandel L, Waltzer L. Characterization of the Drosophila Adult Hematopoietic System Reveals a Rare Cell Population With Differentiation and Proliferation Potential. *Frontiers in cell and developmental biology* **9**, 739357 (2021).

REVIEWERS' COMMENTS

Reviewer #1 (Remarks to the Author):

The authors have addressed my comments.

Reviewer #2 (Remarks to the Author):

The authors addressed my comments

Reviewer #3 (Remarks to the Author):

The manuscript by Liu and colleagues has improved a lot during the revision process, and most of my concerns have been properly addressed. I only have two (minor) comments/suggestions related to the answer to previous comment 6:

1. While it is true that Hml-Gal4 does not express in classical progenitor cells, overexpressions using this driver can still induce changes in hemocyte numbers and/or lamellocyte transdifferentiation.
2. The established assay for blood cell type analysis is staining with plasmatocyte, lamellocyte, and crystal cell specific monoclonal antibodies (these were developed by Istvan Ando's lab in Szeged, Hungary, and widely used by the Drosophila community). Probably even a phalloidin staining would have been more informative than Giemsa.

Reviewer #1:

The authors have addressed my comments.

We are glad that the reviewer is satisfied with our revision. Thanks again for helping us to improve our manuscript.

Reviewer #2:

The authors addressed my comments.

We are glad that the reviewer is satisfied with our revision. Thanks again for helping us to improve our manuscript.

Reviewer #3:

The manuscript by Liu and colleagues has improved a lot during the revision process, and most of my concerns have been properly addressed. I only have two (minor) comments/suggestions related to the answer to previous comment 6:

We are glad that the reviewer is satisfied with most of our revision. Thanks again for helping us to improve our manuscript.

1. While it is true that Hml-Gal4 does not express in classical progenitor cells, overexpressions using this driver can still induce changes in hemocyte numbers and/or lamellocyte transdifferentiation.

Thanks for providing this important information. Nonetheless, the results in our last revision clearly showed comparable number of plasmatocyte, crystal cell and lamellocyte (Fig. S5), as well as comparable total hemocyte number (Fig. 6F) in the flies related to Figs. 5 and 6. These results ruled out the possibility that the observed differential phagocytic ability in Fig. 5 or varied innate immune memory phenotype in Fig. 6 was due to the change of the hemocyte identity or number.

2. The established assay for blood cell type analysis is staining with plasmatocyte, lamellocyte, and crystal cell specific monoclonal antibodies (these were developed by Istvan Ando's lab in Szeged, Hungary, and widely used by the Drosophila community). Probably even a phalloidin staining would have been more informative than Giemsa.

Giemsa staining is an efficient and widely-used method to observe differential hemocytes in diverse insects^{1, 2, 3, 4}. We thank the reviewer for recommending two more methods of *Drosophila* blood cell type analysis. However, these two methods will take us quite a long time to obtain essential reagents and prepare flies. We hope these explanations could address the reviewer's comments to his/her satisfaction.

References

1. Rajak P, Dutta M, Roy S. Altered differential hemocyte count in 3rd instar larvae of *Drosophila melanogaster* as a response to chronic exposure of Acephate. *Interdisciplinary toxicology* **8**, 84-88 (2015).
2. Yu Y, Cao Y, Xia Y, Liu F. Wright-Giemsa staining to observe phagocytes in *Locusta migratoria* infected with *Metarhizium acridum*. *Journal of invertebrate pathology* **139**, 19-24 (2016).
3. Okazaki T, Okudaira N, Iwabuchi K, Fugo H, Nagai T. Apoptosis and adhesion of hemocytes during molting stage of silkworm, *Bombyx mori*. *Zoological science* **23**, 299-304 (2006).
4. Cheng L, Liu WL, Su MP, Huang SC, Wang JR, Chen CH. Prohemocytes are the main cells infected by dengue virus in *Aedes aegypti* and *Aedes albopictus*. *Parasites & vectors* **15**, 137 (2022).